# Constant Stepsize Local GD for Logistic Regression: Acceleration by Instability

**Michael Crawshaw** [1]   **Blake Woodworth** [2]   **Mingrui Liu** [1]

## Abstract

Existing analysis of Local (Stochastic) Gradient Descent for heterogeneous objectives requires stepsizes $\eta \leq 1/K$ where $K$ is the communication interval, which ensures monotonic decrease of the objective. In contrast, we analyze Local Gradient Descent for logistic regression with separable, heterogeneous data using any stepsize $\eta > 0$. With $R$ communication rounds and $M$ clients, we show convergence at a rate $\mathcal{O}(1/\eta KR)$ after an initial unstable phase lasting for $\widetilde{\mathcal{O}}(\eta KM)$ rounds. This improves upon the existing $\mathcal{O}(1/R)$ rate for general smooth, convex objectives. Our analysis parallels the single machine analysis of (Wu et al., 2024a) in which instability is caused by extremely large stepsizes, but in our setting another source of instability is large local updates with heterogeneous objectives.

## 1. Introduction

As the area of distributed optimization grows — owing to recent applications in federated learning (McMahan et al., 2017) and large-scale distributed deep learning (Verbraeken et al., 2020) — the gap between theory and practice has grown proportionally. Local Stochastic Gradient Descent (SGD) and its variants have been successfully used for distributed learning with heterogeneous data in practice for years (Wang et al., 2021; Reddi et al., 2021; Xu et al., 2023), but so far we have little theoretical understanding of this success (Wang et al., 2022).

The majority of theoretical works in distributed optimization take a *worst-case* approach to algorithm analysis: they consider the worst-case efficiency over some large class of optimization problems, such as the class of convex, smooth ob-

jectives satisfying some heterogeneity requirement (Woodworth et al., 2020a;b; Koloskova et al., 2020). While the resulting guarantees are very general, they do not always reflect practice, since they describe the worst-case, rather than cases which may appear in practice. For Local SGD and its deterministic variant, Local GD, these worst-case guarantees rely on the potentially unrealistic condition of small step sizes $\eta \leq \mathcal{O}(1/K)$, where $K$ is the communication interval (Woodworth et al., 2020b; Koloskova et al., 2020). For Local GD, this small step size can guarantee monotonic decrease of the objective, but such stable convergence is far removed from practice, as non-monotonic decrease of the objective is common in practical machine learning (Jastrzebski et al., 2020; Cohen et al., 2021).

Motivated by this gap between theory and practice, we take a problem-specific approach and analyze Local GD for logistic regression. Our central question is:

*** Can Local GD for logistic regression achieve accelerated convergence with a large step size ($\eta \gg 1/K$)?***

Despite the apparent simplicity of this setting, existing theory is unable to answer this question. In the single-machine setting, GD is known to converge for logistic regression with any step size (Wu et al., 2024b;a), and a large enough step size will cause non-monotonic decrease of the objective. For the distributed setting, previous work for this problem considered a two-stage variant of Local GD (Crawshaw et al., 2025), that uses a small step size $\eta \leq \mathcal{O}(1/K)$ before switching to a larger step size later in training. It remains open to analyze the vanilla Local GD with a constant stepsize in this setting.

**Contributions**   In this paper, we prove that Local GD for distributed logistic regression converges with any step size $\eta > 0$ and any communication interval $K \geq 1$. In particular, we show that choosing $\eta K = \widetilde{\Theta}\left(\frac{\gamma^3 R}{M}\right)$ yields a convergence rate faster than existing lower bounds of Local GD for distributed convex optimization (see Section 3 for definitions of all parameters).

Our accelerated convergence crucially uses $\eta K \gg 1$, which violates the condition $\eta \leq \mathcal{O}(1/K)$ from previous work and potentially creates non-monotonic objective decrease across communication rounds. To handle this instability, we adapt

[1]Department of Computer Science, George Mason University, Fairfax, VA, USA [2]Department of Computer Science, George Washington University, Washington, DC, USA. Correspondence to: Michael Crawshaw <mcrawsha@gmu.edu>, Mingrui Liu <mingruil@gmu.edu>.

*Proceedings of the 42$^{nd}$ International Conference on Machine Learning*, Vancouver, Canada. PMLR 267, 2025. Copyright 2025 by the author(s).

Table 1: Upper bounds on the objective gap $F(\boldsymbol{w}) - F_*$ of distributed GD variants for logistic regression, up to constants and logarithmic factors. $R$ is the number of communication rounds, $K$ is the number of local steps, $M$ is the number of clients, and $\gamma$ is the maximum margin of the combined dataset. $(a)$ These bounds are derived in (Crawshaw et al., 2025) by applying the worst-case upper bounds of (Woodworth et al., 2020b) and (Koloskova et al., 2020) to the specific problem of logistic regression. $(b)$ Assuming $R \geq \Omega(Mn\gamma^{-2})$. $(c)$ Assuming $R \geq \widetilde{\Omega}(\max(Mn\gamma^{-2}, KM\gamma^{-3}))$. $(d)$ This lower bound is included for comparison of the rate in terms of $R$ and $K$, and applies to the class of convex, $H$-smooth objectives that have a minimizer $\boldsymbol{w}_*$ with $\|\boldsymbol{w}_*\| \leq B$ and $\|\nabla F_m(\boldsymbol{w}_*) - \nabla F(\boldsymbol{w}_*)\| \leq \zeta_*$. It should be noted that logistic regression with separable data is not a member of this class, because no minimizer $\boldsymbol{w}_*$ exists for this objective.

|  | Step size | Arbitrary $K$ | Best $K$ |
|---|---|---|---|
| Local GD (Woodworth et al., 2020b)[a] | $\eta = \frac{1}{\gamma^{2/3} K R^{1/3}}$ | $\frac{1}{\gamma^2 K R} + \frac{1}{\gamma^{4/3} R^{2/3}}$ | $\frac{1}{\gamma^{4/3} R^{2/3}}$ |
| Local GD (Koloskova et al., 2020)[a] | $\eta = \frac{1}{K}$ | $\frac{1}{\gamma^2 R}$ | $\frac{1}{\gamma^2 R}$ |
| GD (Wu et al., 2024a)[b] | $\eta = \gamma^2 R$ | - | $\frac{1}{\gamma^4 R^2}$ |
| Two-Stage Local GD (Crawshaw et al., 2025) | $\eta_1 = \frac{1}{K}$ $\eta_2 = \min\left(\frac{\gamma^4 R}{KM}, 1\right)$ | $\max\left(\frac{1}{\gamma^2 K R}, \frac{M}{\gamma^6 R^2}\right)$ | $\frac{M}{\gamma^6 R^2}$ |
| Local GD (Corollary 4.3)[c] | $\eta \in \left(1, \frac{\gamma^3 R}{M}\right)$ | $\frac{M}{\gamma^5 R^2}$ | $\frac{M}{\gamma^5 R^2}$ |
| Local GD Lower Bound (Patel et al., 2024)[d] | - | $\frac{HB^2}{R} + \frac{(H\zeta_*^2 B^4)^{1/3}}{R^{2/3}}$ | - |

techniques from the analysis of GD with large step sizes for single-machine logistic regression, introduced by Wu et al. (2024a), which shows that GD operates in an initial unstable phase before entering a stable phase where the objective decreases monotonically. We use these techniques to analyze Local GD by decomposing the algorithm's update into the contribution from each individual data point, and tracking this contribution throughout the local update steps, in order to relate the trajectory of Local GD to that of GD. Consequently, we can show that Local GD also transitions from an unstable phase to a stable phase.

We also experimentally evaluate Local GD for logistic regression with synthetic data and MNIST data, and the results corroborate our theoretical finding that acceleration can be achieved by allowing for non-monotonic objective decrease. To probe the limitations of our theory, we evaluate Local GD under different regimes of $\eta$ and $K$, and accordingly we propose open problems and directions for future research.

**Organization** We first discuss related work (Section 2), then state our problem (Section 3) and give our analysis (Section 4). We provide experimental results (Section 5), then conclude with a discussion of our results and future work (Section 6).

**Notation** For $n \in \mathbb{N}$, we denote $[n] = \{1, \ldots, n\}$. We use $\|\cdot\|$ to denote the $L_2$ norm for vectors and the spectral norm for matrices. Outside of the abstract, we use $\mathcal{O}$, $\Omega$, and $\Theta$ to omit only universal constants. Similarly, $\widetilde{\mathcal{O}}$, $\widetilde{\Omega}$, and $\widetilde{\Theta}$ only omit universal constants and logarithmic terms.

## 2. Related Work

**General Distributed Optimization** Early work in this area focused on distributed algorithms for solving classical learning problems with greater efficiency through parallelization (Mcdonald et al., 2009; McDonald et al., 2010; Zinkevich et al., 2010; Dekel et al., 2012; Balcan et al., 2012; Zhang et al., 2013; Shamir & Srebro, 2014; Arjevani & Shamir, 2015). Recent years have seen a growth of research in distributed optimization due to applications for large-scale distributed training of neural networks (Tang et al., 2020; Verbraeken et al., 2020) and federated learning (McMahan et al., 2017). Federated learning is a paradigm for distributed learning in which user devices collaboratively train a machine learning model without sharing data; see (Kairouz et al., 2021; Wang et al., 2021) for a comprehensive survey.

**Efficiency of Local SGD** Local SGD (also known as Federated Averaging, or FedAvg) is a fundamental algorithm for distributed optimization, both in theory and practice. Convergence guarantees of Local SGD for distributed convex optimization under various conditions were proven by

Stich (2019); Haddadpour & Mahdavi (2019); Woodworth et al. (2020b); Khaled et al. (2020); Koloskova et al. (2020); Glasgow et al. (2022). These works consider the worst-case efficiency of Local SGD for solving large classes of optimization problems, such as the class of problems with smooth, convex objectives with some condition on the heterogeneity between local objectives; we refer to these guarantees as *worst-case baselines*. Lower bounds have established that Local SGD is dominated by Minibatch SGD in the worst case over various problem classes despite the fact that Local SGD tends to outperform Minibatch SGD for practical problems (Woodworth et al., 2020a;b; Glasgow et al., 2022; Patel et al., 2024), and variants of Local SGD remain standard in practice (Wang et al., 2021; 2022; Reddi et al., 2021; Xu et al., 2023). It remains an active topic of research to develop a theoretical understanding of Local SGD and Minibatch SGD that aligns with practical observations (Woodworth et al., 2020b; Glasgow et al., 2022; Wang et al., 2022; Patel et al., 2023; 2024).

**Gradient Methods for Logistic Regression** In this work, we narrow our focus and consider the efficiency of Local GD for solving one particular optimization problem, continuing a line of work which shows that gradient-based optimization algorithms have very particular behavior for certain problems of interest in machine learning. Soudry et al. (2018); Ji & Telgarsky (2019) showed that GD for logistic regression converges to the maximum margin solution without explicit regularization. Gunasekar et al. (2018); Nacson et al. (2019); Ji et al. (2021) proved further implicit regularization results for general steepest descent methods, stochastic gradient descent, and a fast momentum-based algorithm, respectively. A separate line of work observed that GD exhibits non-monotonic decrease in the objective when training neural networks, a phenomenon called the Edge of Stability (Cohen et al., 2021; Damian et al., 2023).

The works which are most closely related to ours are (Wu et al., 2024b;a) and (Crawshaw et al., 2025). Wu et al. (2024b) showed that GD for logistic regression can converge with any positive stepsize, despite non-monotonic decrease of the objective, and that GD converges to the maximum margin solution. Wu et al. (2024a) showed that GD with a large stepsize can achieve accelerated convergence for logistic regression. Crawshaw et al. (2025) proved that a two-stage variant of Local GD can achieve accelerated convergence compared to the worst-case baselines (Koloskova et al., 2020; Woodworth et al., 2020b).

## 3. Problem Setup

We consider a distributed version of binary classification with linearly separable data. The number of clients is denoted by $M$, the number of data points per client as $n$, and

---

**Algorithm 1** Local GD

**Input:** Initialization $\boldsymbol{w}_0 \in \mathbb{R}^d$, rounds $R \in \mathbb{N}$, local steps $K \in \mathbb{N}$, learning rate $\eta > 0$
1: **for** $r = 0, 1, \ldots, R - 1$ **do**
2:    **for** $m \in [M]$ **do**
3:       $\boldsymbol{w}_{r,0}^m \leftarrow \boldsymbol{w}_r$
4:       **for** $k = 0, \ldots, K - 1$ **do**
5:          $\boldsymbol{w}_{r,k+1}^m \leftarrow \boldsymbol{w}_{r,k}^m - \eta \nabla F_m(\boldsymbol{w}_{r,k}^m)$
6:       **end for**
7:    **end for**
8:    $\boldsymbol{w}_{r+1} \leftarrow \frac{1}{M} \sum_{m=1}^M \boldsymbol{w}_{r,K}^m$
9: **end for**

---

the dimension of the input data as $d$. The data consists of $M$ local datasets, one for each client: $D_m = \{(\boldsymbol{x}_i^m, y_i^m)\}_{i \in [n]}$ for each $m \in [M]$, where $\boldsymbol{x}_i^m \in \mathbb{R}^d$ and $y_i^m \in \{-1, 1\}$. We assume that the global dataset $D = \cup_{m \in [M]} D_m$ is linearly separable, that is, there exists some $\boldsymbol{w} \in \mathbb{R}^d$ such that $y \langle \boldsymbol{w}, \boldsymbol{x} \rangle > 0$ for every $(x, y) \in D$. We also denote by $\gamma$ and $\boldsymbol{w}_*$ the maximum margin and the maximum margin classifier for the global dataset, that is,

$$\gamma = \max_{\boldsymbol{w} \in \mathbb{R}^d, \|\boldsymbol{w}\| = 1} \min_{(x,y) \in D} y \langle \boldsymbol{w}, \boldsymbol{x} \rangle \qquad (1)$$

$$\boldsymbol{w}_* = \arg\max_{\boldsymbol{w} \in \mathbb{R}^d, \|\boldsymbol{w}\| = 1} \min_{(x,y) \in D} y \langle \boldsymbol{w}, \boldsymbol{x} \rangle. \qquad (2)$$

Note that $\gamma > 0$ from the assumption of linear separability.

We are interested in studying the behavior of Local Gradient Descent (Algorithm 1) for minimizing the logistic loss of this classification problem. Denoting $\ell(z) = \log(1 + \exp(-z))$, the local objective $F_m : \mathbb{R}^d \to \mathbb{R}$ for client $m \in [M]$ is defined as

$$F_m(\boldsymbol{w}) = \frac{1}{n} \sum_{i=1}^n \ell(y_i^m \langle \boldsymbol{w}, \boldsymbol{x}_i^m \rangle), \qquad (3)$$

and our goal is to approximately solve the following:

$$\min_{\boldsymbol{w} \in \mathbb{R}^d} \left\{ F(\boldsymbol{w}) := \frac{1}{M} \sum_{m=1}^M F_m(\boldsymbol{w}) \right\}. \qquad (4)$$

In this work, we focus on minimization of this training loss, and guarantees for the population loss can be derived using standard techniques.

Notice that the objective depends on each data point $(\boldsymbol{x}_i^m, y_i^m)$ only through the product $y_i^m \boldsymbol{x}_i^m$. Therefore, we can assume without loss of generality that $y_i^m = 1$ for every $m \in [M], i \in [n]$, since we can replace any data point $(\boldsymbol{x}_i^m, -1)$ with $(-\boldsymbol{x}_i^m, 1)$, which preserves the product $y_i^m \boldsymbol{x}_i^m$ and therefore does not change the trajectory of Local GD. We also assume that $\|\boldsymbol{x}_i^m\| \leq 1$ for every $m, i$, which can always be enforced by rescaling all data points by

$\max_{m,i} \|\boldsymbol{x}_i^m\|$. Lastly, we will denote by $H$ the smoothness constant of $F$, that is, $H := \sup_{\boldsymbol{w} \in \mathbb{R}^d} \|\nabla^2 F(\boldsymbol{w})\|$, which satisfies $H \leq 1/4$ when $\|\boldsymbol{x}_i^m\| \leq 1$ (Crawshaw et al., 2025).

## 4. Convergence Analysis

We present two convergence results of Local GD for the logistic regression problem stated in Equation 4. Our Theorem 4.1 gives an upper bound on the average objective $\frac{1}{r} \sum_{s=0}^{r-1} F(\boldsymbol{w}_r)$ over the first $r$ communication rounds, which holds for any $r$. On the other hand, Theorem 4.2 provides a last-iterate upper bound on the objective $F(\boldsymbol{w}_r)$ for every $r$ after a transition time $\tau$. Both of these results hold for any learning rate $\eta > 0$ and any number of local steps $K$. Corollary 4.3 summarizes our results by deriving the error with the best choices of $\eta$ and $K$ for a given communication budget $R$. We first state and discuss the results in Section 4.1, then give an overview of the proofs in Section 4.2. The complete proofs are deferred to Appendix A.

### 4.1. Statement of Results

Theorems 4.1 and 4.2 provide guarantees in two phases: the initial unstable phase (lasting for $\tau$ rounds), and the latter stable phase. During the unstable phase, we cannot provide a last-iterate guarantee, but we can upper bound the average loss over the trajectory. After the loss becomes sufficiently small, Local GD enters the stable phase, where the loss decreases monotonically at every round. These two phases mimic the observed behavior of Local GD in experiments (see Section 5), and align with the behavior of single-machine GD (Wu et al., 2024a).

**Theorem 4.1.** *For every $r \geq 0$, Local GD satisfies*

$$\frac{1}{r} \sum_{s=0}^{r-1} F(\boldsymbol{w}_s) \leq$$
$$26 \frac{\|\boldsymbol{w}_0\|^2 + 1 + \log^2(K + \eta K \gamma^2 r) + \eta^2 K^2}{\eta \gamma^4 r}. \quad (5)$$

Notice that the RHS of Equation 5 grows at most linearly with $\eta$ and quadratically with $K$: this aligns with the intuition that large stepsizes and/or long communication intervals can create instability. Indeed, even if $\eta \leq 1/H$, so that the local objectives are guaranteed to decrease with each local step, the global objective may not decrease monotonically over rounds when $K$ is large, due to a large effective per-round step size $\eta K$. However, for any fixed $\eta$ and $K$, Theorem 4.1 shows that the average loss can be made arbitrarily small with large enough $r$. After at most $\tau$ rounds, $F(\boldsymbol{w}_r)$ will decrease below a certain threshold, after which the global objective will decrease monotonically with each communication round, leading to the following last-iterate guarantee.

**Theorem 4.2.** *Denote $\psi = \min\left(\frac{\gamma}{140\eta KM}, \frac{1}{2Mn}\right)$ and*

$$\tau = \frac{4\gamma\|\boldsymbol{w}_0\| + 2\sqrt{2} + 2\eta + \log\left(1 + \frac{\sqrt{K}}{\sqrt{\eta}\gamma\psi}\right)}{\eta\gamma^2\psi}. \quad (6)$$

*For every $r \geq \tau$, Local GD satisfies*

$$F(\boldsymbol{w}_r) \leq \frac{16}{\eta\gamma^2 K(r - \tau)}. \quad (7)$$

Note that Theorems 4.1 and 4.2 apply for any choice of the stepsize $\eta$ and number of local steps $K$. In contrast with the worst-case analysis which requires that $\eta \leq \mathcal{O}\left(\frac{1}{K}\right)$, ours is the first result showing that Local GD can converge for logistic regression without any restrictions on $\eta$ and $K$. The following corollary shows that, by tuning $\eta$ and $K$, we can achieve an accelerated rate with $R^{-2}$ dependence on $R$, which improves upon the lower bounds of Local GD for general distributed convex optimization (see Table 1).

**Corollary 4.3.** *Suppose $R \geq \widetilde{\Omega}\left(\max\left(\frac{Mn}{\gamma^2}, \frac{KM}{\gamma^3}\right)\right)$. With $\boldsymbol{w}_0 = \boldsymbol{0}$, $\eta \geq 1$, and $\eta K = \widetilde{\Theta}\left(\frac{\gamma^3 R}{M}\right)$, Local GD satisfies*

$$F(\boldsymbol{w}_R) \leq \widetilde{\mathcal{O}}\left(\frac{M}{\gamma^5 R^2}\right). \quad (8)$$

The condition $R \geq \widetilde{\Omega}\left(\max\left(\frac{Mn}{\gamma^2}, \frac{MK}{\gamma^3}\right)\right)$ ensures that $R \geq \tau$, so that training will actually enter the stable phase and decrease the objective at the rate $1/(\eta\gamma^2 KR)$. A similar condition is used in the analysis of GD with large stepsizes for single-machine logistic regression (Wu et al., 2024a).

Also, note that aside from the condition $\eta \geq 1$, the stepsize $\eta$ and the communication interval $K$ always appear together as the product $\eta K$. This means that our guarantee does not distinguish the performance of Local GD as $K$ changes, so long as the stepsize changes to keep $\eta K$ constant. Therefore, it remains open to show whether or not Local GD can actually benefit from the use of local steps for this problem. Indeed, the analysis of GD for single-machine logistic regression (Wu et al., 2024a) immediately implies that for our distributed problem, GD (parallelized over $M$ machines) achieves error $\widetilde{\mathcal{O}}(1/(\gamma^4 R^2))$, which improves upon our guarantee for Local GD in terms of $M$ and $1/\gamma$. We further discuss this comparison in Section 6.

### 4.2. Proof Overview

Throughout the analysis, we will denote $b_{r,i}^m = \langle \boldsymbol{w}_r, \boldsymbol{x}_i^m \rangle$, so that $F_m(\boldsymbol{w}_r) = \frac{1}{n} \sum_{i=1}^{n} \ell(b_{r,i}^m)$. Similarly, we will denote $b_{r,i,k}^m = \langle \boldsymbol{w}_{r,k}^m, \boldsymbol{x}_i^m \rangle$.

The proofs of Theorems 4.1 and 4.2 adapt existing tools introduced by (Wu et al., 2024a) and (Crawshaw et al.,

2025); our application of these tools for our setting relies on a comparison between the trajectories of GD and Local GD by decomposing updates into the contribution from each individual data point $\boldsymbol{x}_i^m$. Specifically, a single GD update starting from $\boldsymbol{w}_r$ is

$$-\eta \nabla F(\boldsymbol{w}_r) = \frac{\eta}{Mn} \sum_{m=1}^{M} \sum_{i=1}^{n} |\ell'(b_{r,i}^m)| \boldsymbol{x}_i^m. \qquad (9)$$

Denoting

$$\beta_{r,i}^m = \frac{\frac{1}{K} \sum_{k=0}^{K-1} |\ell'(b_{r,i,k}^m)|}{|\ell'(b_{r,i}^m)|}, \qquad (10)$$

a single round update of Local GD from $\boldsymbol{w}_r$ can be rewritten

$$\boldsymbol{w}_{r+1} - \boldsymbol{w}_r = -\frac{\eta}{M} \sum_{m=1}^{M} \sum_{k=0}^{K-1} \nabla F_m(\boldsymbol{w}_{r,k}^m) \qquad (11)$$

$$= \frac{\eta K}{Mn} \sum_{m=1}^{M} \sum_{i=1}^{n} \beta_{r,i}^m |\ell'(b_{r,i}^m)| \boldsymbol{x}_i^m. \qquad (12)$$

Comparing Equation 9 and Equation 12, the updates for GD and Local GD can both be represented as linear combinations of the data $\boldsymbol{x}_i^m$, and the two trajectories can be compared by analyzing the coefficients $\beta_{r,i}^m$. By upper and lower bounding $\beta_{r,i}^m$, we can adapt the split comparator and gradient potential techniques of Wu et al. (2024a) (which were introduced for GD) to analyze Local GD during the unstable phase and show a transition to stability.

For the stable phase, we leverage the relationship between the derivatives of the objective function, namely that

$$\|\nabla^2 F(\boldsymbol{w})\| \leq F(\boldsymbol{w}) \quad \text{and} \quad \|\nabla F(\boldsymbol{w})\| \leq F(\boldsymbol{w}), \quad (13)$$

to show that a small objective value $F(\boldsymbol{w})$ implies a small local smoothness $\|\nabla^2 F(\boldsymbol{w}')\|$ for $\|\boldsymbol{w}' - \boldsymbol{w}\| \leq 1$, and this in turn implies monotonic decrease of the objective. A similar argument was used by Crawshaw et al. (2025), but here we use a refined version that allows for any $\eta > 0$, whereas the analysis of Crawshaw et al. (2025) requires $\eta \leq 1/H$.

Below we state key lemmas to sketch the proofs of each theorem, and full proofs are deferred to Appendix A.

**Unstable Phase** As previously mentioned, we aim to apply the split comparator technique of Wu et al. (2024a) to analyze Local GD, and we can do so if we upper and lower bound $\beta_{r,i}^m$. Our lower bound is surprisingly simple:

$$\beta_{r,i}^m = \frac{\frac{1}{K} \sum_{k=0}^{K-1} |\ell'(b_{r,i,k}^m)|}{|\ell'(b_{r,i}^m)|} \geq \frac{1}{K}, \qquad (14)$$

where the inequality simply ignores all terms of the sum in the numerator, except that corresponding to $k = 0$. While this may appear very loose, it is not hard to show in special

cases that this bound is tight up to logarithmic factors for certain values of $\boldsymbol{w}_r$ (see Lemma B.7).

We upper bound $\beta_{r,i}^m$ as

$$\beta_{r,i}^m = \frac{1}{K} \sum_{k=0}^{K-1} \frac{1 + \exp(b_{r,i}^m)}{1 + \exp(b_{r,i,k}^m)} \qquad (15)$$

$$\leq 1 + \exp(b_{r,i}^m) = 1 + \exp(\langle \boldsymbol{w}_r, \boldsymbol{x}_i^m \rangle) \qquad (16)$$

$$\leq 1 + \exp(\|\boldsymbol{w}_r\|), \qquad (17)$$

where the last line uses $\|\boldsymbol{x}_i^m\| \leq 1$. To bound $\|\boldsymbol{w}_r\|$, we apply the split comparator technique of Wu et al. (2024a) to analyze the local trajectories of each round $\{\boldsymbol{w}_{s,k}^m\}_k$, then use this to establish a recursive bound on $\|\boldsymbol{w}_s - \boldsymbol{u}\|$ over rounds, where $\boldsymbol{u} = \boldsymbol{u}_1 + \boldsymbol{u}_2$ is a yet unspecified comparator. The analysis within each round implies that

$$\frac{\|\boldsymbol{w}_{s,K}^m - \boldsymbol{u}\|^2}{2\eta K} + \frac{1}{K} \sum_{k=0}^{K-1} F_m(\boldsymbol{w}_{s,k}^m) \leq$$
$$\frac{\|\boldsymbol{w}_s - \boldsymbol{u}\|^2}{2\eta K} + F_m(\boldsymbol{u}_1), \qquad (18)$$

and in particular that

$$\|\boldsymbol{w}_{s,K}^m - \boldsymbol{u}\| \leq \|\boldsymbol{w}_s - \boldsymbol{u}\| + \sqrt{2\eta K F_m(\boldsymbol{u}_1)}. \qquad (19)$$

Averaging over $m \in [M]$ and recursing over $s \in \{0, \ldots, r-1\}$ implies that

$$\|\boldsymbol{w}_r - \boldsymbol{u}\| \leq \|\boldsymbol{w}_0 - \boldsymbol{u}\| + r\sqrt{2\eta K F(\boldsymbol{u}_1)}, \qquad (20)$$

so

$$\|\boldsymbol{w}_r\| \leq \|\boldsymbol{w}_0\| + 2\|\boldsymbol{u}\| + r\sqrt{2\eta K F(\boldsymbol{u}_1)}. \qquad (21)$$

By choosing $\boldsymbol{u}$ to balance the last two terms on the RHS, we arrive at the following bound.

**Lemma 4.4.** *For every $r \geq 0$,*

$$\|\boldsymbol{w}_r\| \leq \|\boldsymbol{w}_0\| + \frac{\sqrt{2} + \eta + \log(1 + \eta\gamma^2 K r^2)}{\gamma}. \qquad (22)$$

We can now plug this in to Equation 17 to upper bound $\beta_{r,i}^m$. Although the bound for $\beta_{r,i}^m$ is exponential in $\|\boldsymbol{w}_r\|$, Lemma 4.4 shows that $\|\boldsymbol{w}_r\|$ is only logarithmic in $r$, so the resulting upper bound of $\beta_{r,i}^m$ is only polynomial in $r$.

With upper and lower bounds of $\beta_{r,i}^m$, the split comparator technique can be used to analyze Local GD similarly as for GD. The full proof can be found in Appendix A.1.

**Stable Phase** Our error bound for the stable phase uses the following modified descent inequality:

**Lemma 4.5.** *For $\boldsymbol{w}, \boldsymbol{w}' \in \mathbb{R}^d$, if $\|\boldsymbol{w}' - \boldsymbol{w}\| \leq 1$, then for every $m \in [M]$,*

$$F_m(\boldsymbol{w}') - F_m(\boldsymbol{w}) \leq \qquad (23)$$
$$F_m(\boldsymbol{w}) + \langle \nabla F_m(\boldsymbol{w}), \boldsymbol{w}' - \boldsymbol{w} \rangle + 4F_m(\boldsymbol{w}) \|\boldsymbol{w}' - \boldsymbol{w}\|^2.$$

The above descent inequality is proven by using the facts that $\|\nabla^2 F_m(\boldsymbol{w})\| \leq F_m(\boldsymbol{w})$ (Lemma B.1), and $\|\boldsymbol{w}' - \boldsymbol{w}\| \leq \mathcal{O}(1)$ implies that $\|\nabla^2 F_m(\boldsymbol{w}')\| \leq \mathcal{O}(\|\nabla^2 F_m(\boldsymbol{w})\|)$ (Lemma B.3). This descent inequality captures a desirable property of the logistic loss: the local smoothness constant decreases with the objective value, so that large stepsizes can yield monotonic objective decrease as long as the objective is below some threshold.

To use this lemma to bound the error of Local GD, we need to do three things: (1) show that $\|\boldsymbol{w}_{r+1} - \boldsymbol{w}_r\| \leq 1$ when $F(\boldsymbol{w}_r)$ is below some threshold; (2) show that the bias in the update direction $\boldsymbol{w}_{r+1} - \boldsymbol{w}_r$ compared to $-\eta K \nabla F(\boldsymbol{w}_r)$ is negligible when $F(\boldsymbol{w}_r)$ is below some threshold; (3) show that $F(\boldsymbol{w}_r)$ becomes smaller than our desired threshold within $\tau$ rounds.

First, to show that $\|\boldsymbol{w}_{r+1} - \boldsymbol{w}_r\| \leq 1$ based on the magnitude of $F(\boldsymbol{w}_r)$, notice

$$\|\boldsymbol{w}_{r+1} - \boldsymbol{w}_r\| = \eta \left\| \frac{1}{M} \sum_{m=1}^{M} \sum_{k=0}^{K-1} \nabla F_m(\boldsymbol{w}_{r,k}^m) \right\| \quad (24)$$

$$\leq \frac{\eta}{M} \sum_{m=1}^{M} \sum_{k=0}^{K-1} \|\nabla F_m(\boldsymbol{w}_{r,k}^m)\|. \quad (25)$$

We know $\|\nabla F_m(\boldsymbol{w}_{r,k}^m)\| \leq F_m(\boldsymbol{w}_{r,k}^m)$ (Lemma B.1), and if we knew that local updates monotonically decrease the local loss, we further have $F_m(\boldsymbol{w}_{r,k}^m) \leq F_m(\boldsymbol{w}_r)$. Combined with Equation 25, this would yield

$$\|\boldsymbol{w}_{r+1} - \boldsymbol{w}_r\| \leq \eta K F(\boldsymbol{w}_r). \quad (26)$$

In fact, we can use Lemma 4.5 to show that local updates monotonically decrease the local objective, that is, $F_m(\boldsymbol{w}_{r,k+1}^m) \leq F_m(\boldsymbol{w}_{r,k}^m)$, whenever $F_m(\boldsymbol{w}_{r,k}^m) \leq 1/(4\eta)$. This shows that local objectives monotonically decrease across local steps (Lemma 4.6), and this in turn implies that $\|\boldsymbol{w}_{r,k}^m - \boldsymbol{w}_r\| \leq 1$ (Lemma 4.7).

**Lemma 4.6.** *If* $F(\boldsymbol{w}_r) \leq 1/(4\eta M)$ *for some* $r \geq 0$, *then* $F_m(\boldsymbol{w}_{r,k}^m)$ *is decreasing in* $k$ *for every* $m \in [M]$.

**Lemma 4.7.** *If* $F(\boldsymbol{w}_r) \leq 1/(\eta K M)$ *for some* $r \geq 0$, *then* $\|\boldsymbol{w}_{r,k}^m - \boldsymbol{w}_r\| \leq 1$ *for every* $m \in [M], k \in [K]$.

By choosing $k = K$ and averaging over $m \in [M]$, Lemma 4.7 implies that $\|\boldsymbol{w}_{r+1} - \boldsymbol{w}_r\| \leq 1$.

Next, to handle the bias of the update direction, we rewrite the update as

$$\boldsymbol{w}_{r+1} - \boldsymbol{w}_r = -\eta K(\nabla F(\boldsymbol{w}_r) + \boldsymbol{b}_r), \quad (27)$$

where

$$\boldsymbol{b}_r = \frac{1}{MK} \sum_{m=1}^{M} \sum_{k=0}^{K-1} (\nabla F_m(\boldsymbol{w}_{r,k}^m) - \nabla F_m(\boldsymbol{w}_r)). \quad (28)$$

We can bound the magnitude of the bias as follows:

$$\|\boldsymbol{b}_r\| \leq \frac{1}{MK} \sum_{m=1}^{M} \sum_{k=0}^{K-1} \|\nabla F_m(\boldsymbol{w}_{r,k}^m) - \nabla F_m(\boldsymbol{w}_r)\|, \quad (29)$$

and denoting $C = \{(1 - t)\boldsymbol{w}_r + t\boldsymbol{w}_{r,k}^m \mid t \in [0, 1]\}$,

$$\|\nabla F_m(\boldsymbol{w}_{r,k}^m) - \nabla F_m(\boldsymbol{w}_r)\| \quad (30)$$

$$\leq \left( \max_{\boldsymbol{w} \in C} \|\nabla^2 F_m(\boldsymbol{w})\| \right) \|\boldsymbol{w}_{r,k}^m - \boldsymbol{w}_r\| \quad (31)$$

$$\leq \left( \max_{\boldsymbol{w} \in C} F_m(\boldsymbol{w}) \right) \|\boldsymbol{w}_{r,k}^m - \boldsymbol{w}_r\| \quad (32)$$

$$\leq \max\left( F_m(\boldsymbol{w}_r), F_m(\boldsymbol{w}_{r,k}^m) \right) \|\boldsymbol{w}_{r,k}^m - \boldsymbol{w}_r\|, \quad (33)$$

where the last two inequalities use $\|\nabla^2 F_m(\boldsymbol{w})\| \leq F_m(\boldsymbol{w})$ (Lemma B.1) and convexity of $F_m$, respectively. Using Lemmas 4.6 and 4.7, we can already bound the two terms of Equation 33 when $F_m(\boldsymbol{w}_r)$ is small, which gives the following.

**Lemma 4.8.** *If* $F(\boldsymbol{w}_r) \leq \gamma/(70\eta KM)$, *then* $\|\boldsymbol{b}_r\| \leq \frac{1}{5}\|\nabla F(\boldsymbol{w}_r)\|$.

Third, we must show that $F(\boldsymbol{w}_r)$ will be sufficiently small for some $r \leq \tau$ in order to satisfy the conditions of Lemmas 4.6, 4.7, and 4.8. To do this, we adapt the gradient potential argument of Wu et al. (2024a), as previously mentioned, by lower bounding $\beta_{r,i}^m$. We use the same bound as in the proof of Theorem 4.1: $\beta_{r,i}^m \geq 1/K$. This allows us to relate the gradient potential of Local GD to that of GD, and combining this with Lemma 4.4 shows that $F(\boldsymbol{w}_r)$ is sufficiently small to enable stable descent after $\tau$ rounds.

**Lemma 4.9.** *There exists some* $r \leq \tau$ *such that* $F(\boldsymbol{w}_r) \leq \frac{\gamma}{70\eta KM}$.

Finally, to prove Theorem 4.2, we can apply Lemma 4.5 for all $r \geq \tau$. Applying Lemma 4.8 to control the bias of the update direction, we obtain

$$F(\boldsymbol{w}_{r+1}) - F(\boldsymbol{w}_r) \leq -\frac{1}{4}\eta K \|\nabla F(\boldsymbol{w}_r)\|^2. \quad (34)$$

Using $\|\nabla F(\boldsymbol{w}_r)\| \geq \frac{\gamma}{2}F(\boldsymbol{w}_r)$ (Lemma B.1), this leads to a recursion over $F(\boldsymbol{w}_r)$, and unrolling back to round $\tau$ gives exactly Equation 7 from Theorem 4.2. The full proof is given in Appendix A.2.

Corollary 4.3, which gives our result stated in Table 1, is proved in Appendix A.3.

### 4.3. Comparison to Single-Machine Case

When $K = 1$ or $M = 1$, the Local GD algorithm reduces to GD. However, our convergence rate of $M/(\gamma^5 R^2)$ does not exactly recover the $1/(\gamma^4 R^2)$ rate of Wu et al. (2024a) in terms of the dataset's margin $\gamma$. Here we provide some

technical details on the origin of this issue and whether it can be removed.

The issue of our $\gamma$ dependence stems from bounding the bias term $\|\boldsymbol{b}_r\|$ in Lemma 4.8. $\boldsymbol{b}_r$ is the difference between the update direction for a round compared to the global gradient at the beginning of that round. Notice that other conditions for entering the stable phase (Lemma 4.6, Lemma 4.7) only require $F(\boldsymbol{w}_r) \leq O(1/(\eta KM))$, whereas Lemma 4.8 requires $F(\boldsymbol{w}_r) \leq O(\gamma/(\eta KM))$. This additional factor of $\gamma$ needed to bound $\|\boldsymbol{b}_r\|$ creates the worse dependence on $\gamma$ compared with the single-machine case. Note that the gradient bias results from taking multiple local steps before averaging, so it does not appear when $K = 1$ or $M = 1$.

Technically, the requirement $F(\boldsymbol{w}_r) \leq O(\gamma/(\eta KM))$ might be weakened, but with a fine-grained analysis of the Local GD trajectory. First, note that the requirement on $F(\boldsymbol{w}_r)$ is used in Equation 114 of Lemma A.5, for the inequality marked $(iv)$. The need for the factor of $\gamma$ arises from the next inequality (marked $(v)$), where we apply $F(\boldsymbol{w}) \leq 2\|\nabla F(\boldsymbol{w})\|/\gamma$ (Lemma B.2). The additional factor of $\gamma$ is needed to cancel out the $1/\gamma$ from Lemma B.2. Now, if we had a stronger bound in Lemma B.2 — say $F(\boldsymbol{w}) \leq \|\nabla F(\boldsymbol{w})\|$ — then we could remove the extra $\gamma$ factor. The bound $F(\boldsymbol{w}) \leq \|\nabla F(\boldsymbol{w})\|$ does not hold for all $\boldsymbol{w}$, but it does hold for some $\boldsymbol{w}$, namely in the case that $\boldsymbol{w} = t\boldsymbol{w}_*$, where $t$ is a large scalar. So we could possibly improve the gamma dependence if we knew that Local GD converges to the max-margin solution, however, this kind of implicit bias of Local GD with large $\eta$ or $K$ is not known; even in the single-machine case the implicit bias of GD for logistic regression is unknown when the step size scales linearly with the number of iterations (Wu et al., 2024a). We consider this implicit bias analysis as an important direction of future research.

## 5. Experiments

We further investigate the behavior of Local GD for logistic regression through experiments, in order to answer the following questions: **Q1:** Can Local GD converge faster by choosing $\eta$ and $K$ large enough to create non-monotonic objective decrease? **Q2:** Do local steps yield faster convergence if we tune $\eta$ after choosing $K$? **Q3:** Do local steps yield faster convergence if we keep $\eta K$ constant? We investigate Q1 to empirically verify our theoretical findings, whereas Q2 and Q3 are meant to probe the limitations of our theory: our guarantee (Corollary 4.3) does not show any benefit of local steps, and we ask whether such a benefit occurs in practice. We further discuss this limitation of our theory in Section 6. Lastly, we provide an additional experiment with synthetic data in Appendix D.2 to evaluate how optimization behavior is affected by heterogeneity among the margins of each client's local dataset.

**Setup** We evaluate Local GD for a synthetic dataset used by Crawshaw et al. (2025) and for a subset of the MNIST dataset with binarized labels, following (Wu et al., 2024a) and (Crawshaw et al., 2025). The synthetic dataset is a simple testbed with $M = 2$ clients and $n = 1$ data point per client. For MNIST, we use a common protocol (Karimireddy et al., 2020; Crawshaw et al., 2025) to partition 1000 MNIST images among $M = 5$ clients with $n = 200$ images each, in a way that induces heterogeneous feature distributions among clients. Note that $H \leq 1/4$ for these datasets. See Appendix C for complete details of each dataset. Additionally, we provide results with the CIFAR-10 dataset in Appendix D.1.

We run Local GD with a wide range of values for the parameters: $\eta \in \{2^{-2}, 2^0, 2^2, 2^4, 2^6, 2^8, 2^{10}\}$ and $K \in \{2^0, 2^2, 2^4, 2^6\}$. Note that the traditional choice of $\eta = 1/H = 2^2$ is in the middle of the search range for $\eta$, so a large number of these experiments fall outside of the scope of conventional theory. All experiments have a communication budget of $R = 2048$ rounds.

**Results** Our investigations of Q1, Q2, and Q3 are shown in Figures 1, 2, and 3. Note that the results for $\eta = 2^{10}$ are not shown because all such trajectories diverged.

The loss curves in Figure 1 show that the final error reached by Local GD is always made smaller when either $\eta$ or $K$ is increased while the other is held fixed, even when such changes create instability. This answers Q1 affirmatively and is consistent with our theory. Unsurprisingly, increases to $\eta$ create higher loss spikes and require more communication rounds to reach stability, which aligns with our theory. More surprising is that increases to $K$ actually preserve or decrease the rounds required to reach stability while also leading to a smaller final loss! This is consistent across both datasets and is stronger than predicted by our theory, since the transition time $\tau$ in Theorem 4.2 is proportional to $\eta K$.

Figure 2 shows that a larger communication interval $K$ can accelerate convergence when $\eta$ is tuned to $K$, which answers Q2 positively. For the synthetic data, larger choices of $K$ do not increase the time to reach stability, but they lead to a smaller final error. In the MNIST case, we see another stabilizing effect of $K$: larger choices of $K$ permit larger choices of $\eta$! Indeed, setting $\eta = 256$ when $K = 1$ or $K = 4$ caused divergence, whereas this choice led to fast (albeit unstable) convergence when $K = 16$ or $K = 64$.

Lastly, since our Theorem 4.2 does not distinguish the error of Local GD when $\eta K$ is constant, Figure 3 evaluates different parameter choices which have a common value of $\eta K$. For both datasets, the final error reached by Local GD is nearly identical for all parameter choices, which leans toward a negative answer for Q3. However, we can see that the number of rounds required to reach the stable phase

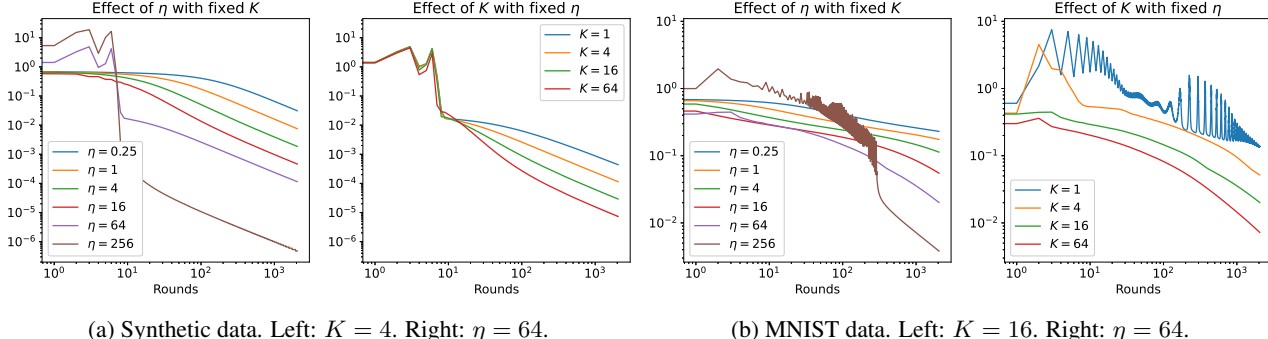

(a) Synthetic data. Left: $K = 4$. Right: $\eta = 64$.       (b) MNIST data. Left: $K = 16$. Right: $\eta = 64$.

Figure 1: Objective gap when varying one of $\eta, K$ and keeping the other fixed. In general, Local GD converges faster when $\eta$ and $K$ are larger, despite the initial instability in early rounds.

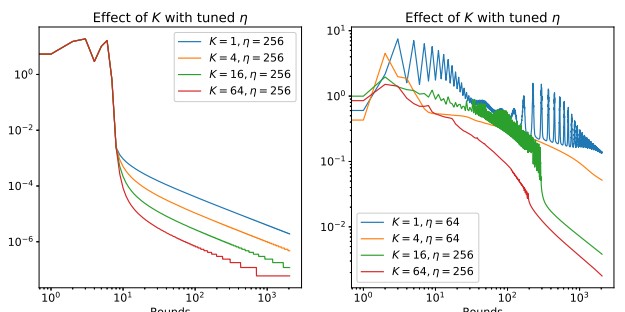

Figure 2: Objective gap for different values of $K$ with tuned $\eta$. Left: Synthetic data. Right: MNIST data.

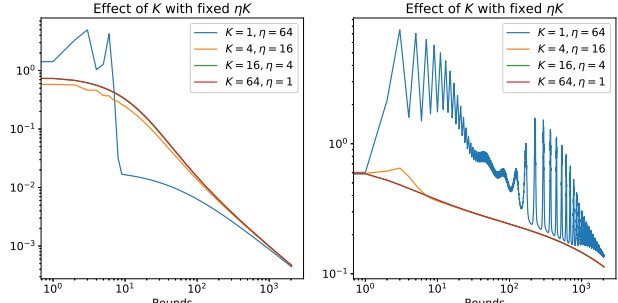

Figure 3: Objective gap for different values of $\eta, K$ with constant $\eta K$. Left: Synthetic data. Right: MNIST data.

tends to decrease as $K$ increases, which still suggests that there may be room for improvement in our bound of the transition time in Theorem 4.2.

Together, our experimental results confirm that instability is an important ingredient for the fast convergence of Local GD for logistic regression. Further, they suggest that Local GD with $K > 1$ may be able to outperform GD under the same communication budget, which is even stronger than our current guarantees. We discuss this possibility as a direction of future research in Section 6.

## 6. Discussion

We have presented the first results showing that Local GD for logistic regression can converge with any step size $\eta > 0$ and any communication interval $K$, and our convergence rate improves upon that guaranteed by the worst-case analysis which is known to be tight (Koloskova et al., 2020; Woodworth et al., 2020b; Patel et al., 2024). Below we discuss the problem-specific approach, limitations of our results, and suggest directions for follow up work.

**Choice of Problem Class** The conventional optimization analysis of distributed learning focuses on providing guarantees of efficiency in the worst-case over large classes of optimization problems. The question is, which class of problems should we analyze? Certain classes of problems lend themselves well to theoretical analysis, such as those satisfying a heterogeneity condition like uniformly bounded gradient dissimilarity (Woodworth et al., 2020b), or bounded gradient dissimilarity at the optimum (Koloskova et al., 2020); however, such conditions have come into question, since they lead to worst-case complexities that do not explain algorithm behavior for practical problems (Wang et al., 2022; Patel et al., 2023; 2024). These works have attempted to find the "right" heterogeneity condition, but so far (to the best of our knowledge), no such condition has explained the significant advantage enjoyed by Local SGD over Minibatch SGD in practice. In this work, by focusing on a specific problem, we investigate the possibility that algorithm performance can be explained according to the specific problem structure rather than general heterogeneity conditions, as discussed by Patel et al. (2024) and Crawshaw et al. (2025). Even though this approach is less

general than the conventional style, we believe that a narrow analysis which accurately describes practice has a different kind of value than a general analysis which does not, and is an important direction for the community to pursue.

**Usefulness of Local Steps**    The main limitation of our results is that our error bound for Local GD is strictly worse than that of GD for $R$ steps (Wu et al., 2024a) in terms of $M$ and $1/\gamma$ (see Table 1). If we are to accept these results, one should set $K = 1$ and parallelize GD over $M$ machines rather than use Local GD with $K > 1$, but it remains open whether our analysis for Local GD can be improved to match (or even dominate) GD. Based on our experiments, we conjecture that Local GD with $K > 1$ can converge faster than GD, and this suggests two open problems: (1) Provide a lower bound of GD for logistic regression, and (2) Determine whether Local GD with $K > 1$ can converge with error smaller than $R^{-\alpha}$ for some $\alpha > 2$. Our current results are insufficient to show any advantage to setting $K > 1$, not only because of the unfavorable comparison with GD, but also because $\eta$ and $K$ appear in our Theorem 4.2 only through the product $\eta K$ (excluding non-dominating terms of the transition time $\tau$). This means that any error guaranteed by choosing stepsize $\eta$ and communication interval $K$ can also be guaranteed with stepsize $\eta K$ and communication interval 1, so that an interval larger than 1 does not produce any advantage. The challenge of proving an advantage from local steps is fundamental in distributed optimization (Woodworth et al., 2020b; Glasgow et al., 2022; Patel et al., 2024), and we hope to address this in future work.

**Future Extensions**    There are several natural extensions of our work, given the narrow focus of the problem setting. Since SGD for logistic regression was analyzed by Wu et al. (2024a) using similar techniques as we have leveraged in this work, one direction is to extend our analysis for Local SGD. These same techniques were applied by Cai et al. (2024) to analyze GD for training two-layer neural networks with approximately homogeneous activations, so another direction is to analyze the distributed training of two-layer networks with Local GD. Lastly, one could attempt to generalize our analysis for a larger class of problems, by formulating some general problem class for which Local GD outperforms the existing worst-case lower bounds. We leave these directions for future work.

## Acknowledgements

Thank you to the anonymous reviewers for the valuable feedback. Michael Crawshaw is supported by the Institute for Digital Innovation fellowship. Mingrui Liu is supported by a ORIEI seed funding, an IDIA P3 fellowship from George Mason University, and NSF awards #2436217, #2425687.

## Impact Statement

This paper presents work whose goal is to advance the field of Machine Learning. There are many potential societal consequences of our work, none which we feel must be specifically highlighted here.

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

# A. Proofs of Main Results

## A.1. Proof of Theorem 4.1

**Lemma A.1** (Restatement of Lemma 4.4). *For every $r \geq 0$,*

$$\|\boldsymbol{w}_r\| \leq \|\boldsymbol{w}_0\| + \frac{\sqrt{2} + \eta + \log(1 + \eta\gamma^2 K r^2)}{\gamma}. \tag{35}$$

*Proof.* Recall that Wu et al. (2024a) introduced a large stepsize analysis of GD for logistic regression, which provides an upper bound on the norm of the parameter at each step. We wish to achieve a similar bound for the norm of the parameter found by Local GD. To accomplish this, we treat the local training of each client during each round as GD on a logistic regression problem, and we apply the "split comparator" technique of Wu et al. (2024a). This leads to a recursive upper bound on the norm of the global parameter $\|\boldsymbol{w}_r\|$, and unrolling yields the desired bound. We demonstrate this argument below.

Let $0 \leq s < r$ and $m \in [M]$. Define $\boldsymbol{u}_1 = \lambda_1 \boldsymbol{w}_*, \boldsymbol{u}_2 = \lambda_2 \boldsymbol{w}_*$, and $\boldsymbol{u} = \boldsymbol{u}_1 + \boldsymbol{u}_2$, where $\lambda_1, \lambda_2$ will be chosen later and will not depend on $s$ or $m$. Note that $\boldsymbol{u}$ is a scalar multiple of $\boldsymbol{w}_*$, which is the maximum margin predictor of the global dataset, not that of any local dataset. We start by applying the split comparator technique of (Wu et al., 2024a) to the local updates of client $m$ at round $s$, which takes $K$ gradient steps with learning rate $\eta$ on the objective $F_m$, initialized from $\boldsymbol{w}_s$. For every $0 \leq k < K$,

$$\|\boldsymbol{w}_{s,k+1}^m - \boldsymbol{u}\|^2 = \|(\boldsymbol{w}_{s,k}^m - \boldsymbol{u}) + (\boldsymbol{w}_{s,k+1}^m - \boldsymbol{w}_{s,k}^m)\|^2 \tag{36}$$

$$= \|\boldsymbol{w}_{s,k}^m - \boldsymbol{u}\|^2 + 2 \langle \boldsymbol{w}_{s,k+1}^m - \boldsymbol{w}_{s,k}^m, \boldsymbol{w}_{s,k}^m - \boldsymbol{u} \rangle + \|\boldsymbol{w}_{s,k+1}^m - \boldsymbol{w}_{s,k}^m\|^2 \tag{37}$$

$$= \|\boldsymbol{w}_{s,k}^m - \boldsymbol{u}\|^2 + 2\eta \langle \nabla F_m(\boldsymbol{w}_{s,k}^m), \boldsymbol{u} - \boldsymbol{w}_{s,k}^m \rangle + \eta^2 \|\nabla F_m(\boldsymbol{w}_{s,k}^m)\|^2 \tag{38}$$

$$= \|\boldsymbol{w}_{s,k}^m - \boldsymbol{u}\|^2 + \underbrace{2\eta \langle \nabla F_m(\boldsymbol{w}_{s,k}^m), \boldsymbol{u}_1 - \boldsymbol{w}_{s,k}^m \rangle}_{A_1} \tag{39}$$

$$+ \underbrace{2\eta \langle \nabla F_m(\boldsymbol{w}_{s,k}^m), \boldsymbol{u}_2 \rangle + \eta^2 \|\nabla F_m(\boldsymbol{w}_{s,k}^m)\|^2}_{A_2} \tag{40}$$

The first term $A_1$ is easily bounded by convexity of $F_m$:

$$A_1 = 2\eta \langle \nabla F_m(\boldsymbol{w}_{s,k}^m), \boldsymbol{u}_1 - \boldsymbol{w}_{s,k}^m \rangle \leq 2\eta(F_m(\boldsymbol{u}_1) - F_m(\boldsymbol{w}_{s,k}^m)). \tag{41}$$

The second term $A_2$ can be bounded by the Lipschitz property of $F_m$ together with a choice of $\boldsymbol{u}_2$:

$$A_2 = \eta \left( 2 \langle \nabla F_m(\boldsymbol{w}_{s,k}^m), \boldsymbol{u}_2 \rangle + \eta \|\nabla F_m(\boldsymbol{w}_{s,k}^m)\|^2 \right) \tag{42}$$

$$\overset{(i)}{=} \eta \left( -\frac{2}{n} \sum_{i=1}^n \frac{\langle \boldsymbol{x}_i^m, \boldsymbol{u}_2 \rangle}{1 + \exp(\langle \boldsymbol{w}_{s,k}^m, \boldsymbol{x}_i^m \rangle)} + \eta \left\| \frac{1}{n} \sum_{i=1}^n \frac{\boldsymbol{x}_i^m}{1 + \exp(\langle \boldsymbol{w}_{s,k}^m, \boldsymbol{x}_i^m \rangle)} \right\|^2 \right) \tag{43}$$

$$\overset{(ii)}{\leq} \eta \left( -\frac{2\lambda_2}{n} \sum_{i=1}^n \frac{\langle \boldsymbol{x}_i^m, \boldsymbol{w}_* \rangle}{1 + \exp(\langle \boldsymbol{w}_{s,k}^m, \boldsymbol{x}_i^m \rangle)} + \frac{\eta}{n} \sum_{i=1}^n \left\| \frac{\boldsymbol{x}_i^m}{1 + \exp(\langle \boldsymbol{w}_{s,k}^m, \boldsymbol{x}_i^m \rangle)} \right\|^2 \right) \tag{44}$$

$$\overset{(iii)}{\leq} \eta \left( -\frac{2\gamma\lambda_2}{n} \sum_{i=1}^n \frac{1}{1 + \exp(\langle \boldsymbol{w}_{s,k}^m, \boldsymbol{x}_i^m \rangle)} + \frac{\eta}{n} \sum_{i=1}^n \left\| \frac{\boldsymbol{x}_i^m}{1 + \exp(\langle \boldsymbol{w}_{s,k}^m, \boldsymbol{x}_i^m \rangle)} \right\| \right) \tag{45}$$

$$\overset{(iv)}{\leq} \eta \left( -\frac{2\gamma\lambda_2}{n} \sum_{i=1}^n \frac{1}{1 + \exp(\langle \boldsymbol{w}_{s,k}^m, \boldsymbol{x}_i^m \rangle)} + \frac{\eta}{n} \sum_{i=1}^n \frac{1}{1 + \exp(\langle \boldsymbol{w}_{s,k}^m, \boldsymbol{x}_i^m \rangle)} \right) \tag{46}$$

$$= \frac{\eta}{n} \sum_{i=1}^n \frac{-2\gamma\lambda_2 + \eta}{1 + \exp(\langle \boldsymbol{w}_{s,k}^m, \boldsymbol{x}_i^m \rangle)}, \tag{47}$$

where $(i)$ uses the definition of $\nabla F_m$, $(ii)$ uses the definition of $\boldsymbol{u}_2$ and Jensen's inequality, and both $(iii)$ and $(iv)$ use

$\|\boldsymbol{x}_i^m\| \leq 1$. Therefore, choosing $\lambda_2 = \eta/(2\gamma)$ implies that $A_2 \leq 0$. Plugging back to Equation 40,

$$\|\boldsymbol{w}_{s,k+1}^m - \boldsymbol{u}\|^2 \leq \|\boldsymbol{w}_{s,k}^m - \boldsymbol{u}\|^2 + 2\eta(F_m(\boldsymbol{u}_1) - F_m(\boldsymbol{w}_{s,k}^m)) \tag{48}$$

$$F_m(\boldsymbol{w}_{s,k}^m) \leq \frac{\|\boldsymbol{w}_{s,k}^m - \boldsymbol{u}\|^2 - \|\boldsymbol{w}_{s,k+1}^m - \boldsymbol{u}\|^2}{2\eta} + F_m(\boldsymbol{u}_1). \tag{49}$$

Averaging over $k \in \{0, \ldots, K-1\}$,

$$\frac{1}{K} \sum_{k=0}^{K-1} F_m(\boldsymbol{w}_{s,k}^m) \leq \frac{\|\boldsymbol{w}_s - \boldsymbol{u}\|^2 - \|\boldsymbol{w}_{s,K}^m - \boldsymbol{u}\|^2}{2\eta K} + F_m(\boldsymbol{u}_1) \tag{50}$$

$$\frac{\|\boldsymbol{w}_{s,K}^m - \boldsymbol{u}\|^2}{2\eta K} + \frac{1}{K} \sum_{k=0}^{K-1} F_m(\boldsymbol{w}_{s,k}^m) \leq \frac{\|\boldsymbol{w}_s - \boldsymbol{u}\|^2}{2\eta K} + F_m(\boldsymbol{u}_1). \tag{51}$$

In particular, this implies

$$\frac{\|\boldsymbol{w}_{s,K}^m - \boldsymbol{u}\|^2}{2\eta K} \leq \frac{\|\boldsymbol{w}_s - \boldsymbol{u}\|^2}{2\eta K} + F_m(\boldsymbol{u}_1), \tag{52}$$

so

$$\|\boldsymbol{w}_{s,K}^m - \boldsymbol{u}\| \leq \sqrt{\|\boldsymbol{w}_s - \boldsymbol{u}\|^2 + 2\eta K F_m(\boldsymbol{u}_1)} \leq \|\boldsymbol{w}_s - \boldsymbol{u}\| + \sqrt{2\eta K F_m(\boldsymbol{u}_1)}. \tag{53}$$

Recall that $\boldsymbol{w}_{s+1} = \frac{1}{M} \sum_{m=1}^{M} \boldsymbol{w}_{s,K}^m$. So averaging over $m$,

$$\|\boldsymbol{w}_{s+1} - \boldsymbol{u}\| = \left\| \frac{1}{M} \sum_{m=1}^{M} \boldsymbol{w}_{s,k}^m - \boldsymbol{u} \right\| \leq \frac{1}{M} \sum_{m=1}^{M} \|\boldsymbol{w}_{s,k}^m - \boldsymbol{u}\| \tag{54}$$

$$\leq \|\boldsymbol{w}_s - \boldsymbol{u}\| + \frac{1}{M} \sum_{m=1}^{M} \sqrt{2\eta K F_m(\boldsymbol{u}_1)} \tag{55}$$

$$\overset{(i)}{\leq} \|\boldsymbol{w}_s - \boldsymbol{u}\| + \sqrt{2\eta K F(\boldsymbol{u}_1)}, \tag{56}$$

where $(i)$ uses the fact that $\sqrt{\cdot}$ is concave together with Jensen's inequality. We can now unroll this recursion over $s \in \{0, \ldots, r-1\}$ to obtain

$$\|\boldsymbol{w}_r - \boldsymbol{u}\| \leq \|\boldsymbol{w}_0 - \boldsymbol{u}\| + \sqrt{2\eta K r^2 F(\boldsymbol{u}_1)} \leq \|\boldsymbol{w}_0\| + \|\boldsymbol{u}\| + \sqrt{2\eta K r^2 F(\boldsymbol{u}_1)}. \tag{57}$$

so

$$\|\boldsymbol{w}_r\| \leq \|\boldsymbol{w}_r - \boldsymbol{u}\| + \|\boldsymbol{u}\| \leq \|\boldsymbol{w}_0\| + 2\|\boldsymbol{u}\| + \sqrt{2\eta K r^2 F(\boldsymbol{u}_1)} \tag{58}$$

$$= \|\boldsymbol{w}_0\| + 2\lambda_1 + 2\lambda_2 + \sqrt{2\eta K r^2 F(\lambda_1 \boldsymbol{w}_*)}. \tag{59}$$

It only remains to choose $\lambda_1$. Note that

$$F(\lambda_1 \boldsymbol{w}_*) = \frac{1}{Mn} \sum_{m=1}^{M} \sum_{i=1}^{n} \log(1 + \exp(-\lambda_1 \langle \boldsymbol{w}_*, \boldsymbol{x}_i^m \rangle)) \tag{60}$$

$$\overset{(i)}{\leq} \frac{1}{Mn} \sum_{m=1}^{M} \sum_{i=1}^{n} \exp(-\lambda_1 \langle \boldsymbol{w}_*, \boldsymbol{x}_i^m \rangle) \tag{61}$$

$$\overset{(ii)}{\leq} \exp(-\lambda_1 \gamma), \tag{62}$$

where $(i)$ uses $\log(1+x) \leq x$ for $x \geq 0$ and $(ii)$ uses the definition of $\boldsymbol{w}_*$. Therefore, choosing $\lambda_1 = \frac{1}{\gamma} \log(1 + \eta \gamma^2 K r^2)$ yields

$$F(\lambda_1 \boldsymbol{w}_*) \leq \frac{1}{1 + \eta \gamma^2 K r^2} \leq \frac{1}{\eta \gamma^2 K r^2}, \tag{63}$$

so

$$\|\boldsymbol{w}_r\| \le \|\boldsymbol{w}_0\| + \frac{2}{\gamma} \log(1 + \eta\gamma^2 K r^2) + \frac{\eta}{\gamma} + \sqrt{2\eta K r^2 \frac{1}{\eta\gamma^2 K r^2}} \tag{64}$$

$$= \|\boldsymbol{w}_0\| + \frac{\sqrt{2} + \eta + \log(1 + \eta\gamma^2 K r^2)}{\gamma}. \tag{65}$$

$\square$

**Theorem A.2** (Restatement of Theorem 4.1). *For every $r \ge 0$, Local GD satisfies*

$$\frac{1}{r} \sum_{s=0}^{r-1} F(\boldsymbol{w}_s) \le 26 \frac{\|\boldsymbol{w}_0\|^2 + 1 + \log^2(K + \eta K\gamma^2 r) + \eta^2 K^2}{\eta\gamma^4 r}. \tag{66}$$

*Proof.* To achieve this bound on the loss of Local GD, we again adapt the split comparator technique of (Wu et al., 2024a). This time, we consider the trajectory of the global model $\boldsymbol{w}_r$, instead of the trajectory of locally updated models $\boldsymbol{w}_{r,k}^m$ as in Lemma 4.4. To apply this technique for Local GD, we have to account for the fact that the update direction $\boldsymbol{w}_{r+1} - \boldsymbol{w}_r$ is not equal to the global gradient $\nabla F(\boldsymbol{w}_r)$. However, both the update direction and the global gradient are linear combinations of the data $\{\boldsymbol{x}_i^m\}_{m,i}$, and we account for the discrepancy between the two by bounding the ratio of their linear combination coefficients.

Let $\boldsymbol{u}_1 = \lambda_1 \boldsymbol{w}_*, \boldsymbol{u}_2 = \lambda_2 \boldsymbol{w}_*$, where $\lambda_1$ and $\lambda_2$ will be determined later, and let $\boldsymbol{u} = \boldsymbol{u}_1 + \boldsymbol{u}_2$. Then

$$\|\boldsymbol{w}_{s+1} - \boldsymbol{u}\|^2 = \|(\boldsymbol{w}_s - \boldsymbol{u}) + (\boldsymbol{w}_{s+1} - \boldsymbol{w}_s)\|^2 \tag{67}$$

$$= \|\boldsymbol{w}_s - \boldsymbol{u}\|^2 + 2\langle \boldsymbol{w}_{s+1} - \boldsymbol{w}_s, \boldsymbol{w}_s - \boldsymbol{u}\rangle + \|\boldsymbol{w}_{s+1} - \boldsymbol{w}_s\|^2 \tag{68}$$

$$= \|\boldsymbol{w}_s - \boldsymbol{u}\|^2 + \frac{2\eta}{M} \sum_{m=1}^{M} \sum_{k=0}^{K-1} \langle \nabla F_m(\boldsymbol{w}_{s,k}^m), \boldsymbol{u} - \boldsymbol{w}_s\rangle + \eta^2 \left\| \frac{1}{M} \sum_{m=1}^{M} \sum_{k=0}^{K-1} \nabla F_m(\boldsymbol{w}_{s,k}^m) \right\|^2 \tag{69}$$

$$= \|\boldsymbol{w}_s - \boldsymbol{u}\|^2 + \underbrace{\frac{2\eta}{M} \sum_{m=1}^{M} \sum_{k=0}^{K-1} \langle \nabla F_m(\boldsymbol{w}_{s,k}^m), \boldsymbol{u}_1 - \boldsymbol{w}_s\rangle}_{A_1} \tag{70}$$

$$+ \underbrace{\frac{2\eta}{M} \sum_{m=1}^{M} \sum_{k=0}^{K-1} \langle \nabla F_m(\boldsymbol{w}_{s,k}^m), \boldsymbol{u}_2\rangle + \eta^2 \left\| \frac{1}{M} \sum_{m=1}^{M} \sum_{k=0}^{K-1} \nabla F_m(\boldsymbol{w}_{s,k}^m) \right\|^2}_{A_2}. \tag{71}$$

To bound $A_1$, we express the local gradient of the local models $\nabla F_m(\boldsymbol{w}_{s,k}^m)$ in terms of the local gradient of the preceding global model $\nabla F_m(\boldsymbol{w}_s)$. For any $\boldsymbol{w}$,

$$\nabla F_m(\boldsymbol{w}) = \frac{1}{n} \sum_{i=1}^{n} \nabla F_{m,i}(\boldsymbol{w}) = \frac{-1}{n} \sum_{i=1}^{n} \frac{\boldsymbol{x}_i^m}{1 + \exp(\langle \boldsymbol{x}_i^m, \boldsymbol{w}\rangle)}. \tag{72}$$

So denoting $\beta_{s,i,k}^m = (1 + \exp(b_{s,i}^m))/(1 + \exp(b_{s,i,k}^m))$ and $F_{m,i}(\boldsymbol{w}) = \log(1 + \exp(-\langle \boldsymbol{w}, \boldsymbol{x}_i^m\rangle))$,

$$\nabla F_m(\boldsymbol{w}_{s,k}^m) = \frac{1}{n} \sum_{i=1}^{n} \frac{-\boldsymbol{x}_i^m}{1 + \exp(b_{s,i,k}^m)} = \frac{1}{n} \sum_{i=1}^{n} \frac{1 + \exp(b_{s,i}^m)}{1 + \exp(b_{s,i,k}^m)} \frac{-\boldsymbol{x}_i^m}{1 + \exp(b_{s,i}^m)} = \frac{1}{n} \sum_{i=1}^{n} \beta_{s,i,k}^m \nabla F_{m,i}(\boldsymbol{w}_s). \tag{73}$$

Notice, from the definition of $\beta_{s,k}^m$,

$$\beta_{s,k}^m := \frac{1}{K} \sum_{k=0}^{K-1} \frac{|\ell'(b_{s,i,k}^m)|}{|\ell'(b_{s,i}^m)|} = \frac{1}{K} \sum_{k=0}^{K-1} \frac{1 + \exp(b_{s,i}^m)}{1 + \exp(b_{s,i,k}^m)} = \frac{1}{K} \sum_{k=0}^{K-1} \beta_{s,i,k}^m, \tag{74}$$

so

$$A_1 = \frac{2\eta}{Mn} \sum_{m=1}^{M} \sum_{k=0}^{K-1} \sum_{i=1}^{n} \beta_{s,i,k}^{m} \langle \nabla F_{m,i}(\boldsymbol{w}_s), \boldsymbol{u}_1 - \boldsymbol{w}_s \rangle \tag{75}$$

$$\overset{(i)}{\leq} \frac{2\eta}{Mn} \sum_{m=1}^{M} \sum_{k=0}^{K-1} \sum_{i=1}^{n} \beta_{s,i,k}^{m} (F_{m,i}(\boldsymbol{u}_1) - F_{m,i}(\boldsymbol{w}_s)) \tag{76}$$

$$= \frac{2\eta K}{Mn} \sum_{m=1}^{M} \sum_{i=1}^{n} \beta_{s,i}^{m} F_{m,i}(\boldsymbol{u}_1) - \frac{2\eta K}{Mn} \sum_{m=1}^{M} \sum_{i=1}^{n} \beta_{s,i}^{m} F_{m,i}(\boldsymbol{w}_s). \tag{77}$$

where $(i)$ uses the convexity of $F_{m,i}$. We can now bound the two terms of Equation 77 with upper and lower bounds of $\beta_{s,i}^{m}$, respectively. Denoting $\phi = \|\boldsymbol{w}_0\| + \frac{\sqrt{2}+\eta+\log(1+\eta\gamma^2 Kr^2)}{\gamma}$,

$$\beta_{s,i}^{m} = \frac{1}{K} \sum_{k=0}^{K-1} \frac{1 + \exp(b_{s,i}^{m})}{1 + \exp(b_{s,i,k}^{m})} \leq 1 + \exp(b_{s,i}^{m}) = 1 + \exp(\langle \boldsymbol{w}_s, \boldsymbol{x}_i^m \rangle) \tag{78}$$

$$\overset{(i)}{\leq} 1 + \exp(\|\boldsymbol{w}_s\|) \overset{(ii)}{\leq} 1 + \exp\left(\|\boldsymbol{w}_0\| + \frac{\sqrt{2} + \eta + \log(1 + \eta\gamma^2 Ks^2)}{\gamma}\right) \tag{79}$$

$$\leq 2\exp(\phi), \tag{80}$$

where $(i)$ uses Cauchy-Schwarz together with $\|\boldsymbol{x}_i^m\| \leq 1$ and $(ii)$ uses Lemma 4.4. Also,

$$\beta_{s,i}^{m} = \frac{1}{K} \sum_{k=0}^{K-1} \frac{1 + \exp(b_{s,i}^{m})}{1 + \exp(b_{s,i,k}^{m})} \geq \frac{1}{K} \frac{1 + \exp(b_{s,i}^{m})}{1 + \exp(b_{s,i,0}^{m})} = \frac{1}{K}. \tag{81}$$

The step $\beta_{s,i}^{m} \geq \frac{1}{K}$ was mentioned in our proof overview, and it will be used again in the proof of Lemma 4.9. See Lemma B.7 for a discussion on the tightness of this bound. Plugging Equation 80 and Equation 81 into Equation 77,

$$A_1 \leq \frac{4\eta K \exp(\phi)}{Mn} \sum_{m=1}^{M} \sum_{i=1}^{n} F_{m,i}(\boldsymbol{u}_1) - \frac{2\eta}{Mn} \sum_{m=1}^{M} \sum_{i=1}^{n} F_{m,i}(\boldsymbol{w}_s) \tag{82}$$

$$\leq 4\eta K \exp(\phi) F(\boldsymbol{u}_1) - 2\eta F(\boldsymbol{w}_s). \tag{83}$$

This bounds $A_1$. For $A_2$,

$$A_2 = \frac{2\eta}{M} \sum_{m=1}^{M} \sum_{k=0}^{K-1} \langle \nabla F_m(\boldsymbol{w}_{s,k}^{m}), \boldsymbol{u}_2 \rangle + \eta^2 K^2 \left\| \frac{1}{MK} \sum_{m=1}^{M} \sum_{k=0}^{K-1} \nabla F_m(\boldsymbol{w}_{s,k}^{m}) \right\|^2 \tag{84}$$

$$\leq \frac{2\eta}{M} \sum_{m=1}^{M} \sum_{k=0}^{K-1} \langle \nabla F_m(\boldsymbol{w}_{s,k}^{m}), \boldsymbol{u}_2 \rangle + \frac{\eta^2 K}{M} \sum_{m=1}^{M} \sum_{k=0}^{K-1} \left\| \nabla F_m(\boldsymbol{w}_{s,k}^{m}) \right\|^2 \tag{85}$$

$$= \frac{\eta}{M} \sum_{m=1}^{M} \sum_{k=0}^{K-1} \left( 2 \langle \nabla F_m(\boldsymbol{w}_{r,k}^{m}), \boldsymbol{u}_2 \rangle + \eta K \left\| \nabla F_m(\boldsymbol{w}_{s,k}^{m}) \right\|^2 \right) \tag{86}$$

$$\overset{(i)}{\leq} \frac{\eta}{M} \sum_{m=1}^{M} \sum_{k=0}^{K-1} \left( 2 \langle \nabla F_m(\boldsymbol{w}_{s,k}^{m}), \boldsymbol{u}_2 \rangle + \eta K \left\| \nabla F_m(\boldsymbol{w}_{s,k}^{m}) \right\| \right) \tag{87}$$

$$= \frac{\eta}{Mn} \sum_{m=1}^{M} \sum_{k=0}^{K-1} \sum_{i=1}^{n} \left( -\frac{2 \langle \boldsymbol{x}_i^m, \boldsymbol{u}_2 \rangle}{1 + \exp(\langle \boldsymbol{x}_i^m, \boldsymbol{w}_{s,k}^{m} \rangle)} + \frac{\eta K \|\boldsymbol{x}_i^m\|}{1 + \exp(\langle \boldsymbol{x}_i^m, \boldsymbol{w}_{s,k}^{m} \rangle)} \right) \tag{88}$$

$$\overset{(ii)}{=} \frac{\eta}{Mn} \sum_{m=1}^{M} \sum_{k=0}^{K-1} \sum_{i=1}^{n} \frac{-2\lambda_2 \langle \boldsymbol{x}_i^m, \boldsymbol{w}_* \rangle + \eta K \|\boldsymbol{x}_i^m\|}{1 + \exp(\langle \boldsymbol{x}_i^m, \boldsymbol{w}_{s,k}^{m} \rangle)} \tag{89}$$

$$\leq \frac{\eta}{Mn} \sum_{m=1}^{M} \sum_{k=0}^{K-1} \sum_{i=1}^{n} \frac{-2\gamma\lambda_2 + \eta K}{1 + \exp(\langle \boldsymbol{x}_i^m, \boldsymbol{w}_{s,k}^{m} \rangle)}, \tag{90}$$

where $(i)$ uses the fact that $\|\nabla F_m(\boldsymbol{w})\| \leq 1$, coming from Equation 72 and $\|\boldsymbol{x}_i^m\| \leq 1$, and $(ii)$ uses the definition of $\boldsymbol{u}_2$. Choosing $\lambda_2 = \eta K/(2\gamma)$ then implies that $A_2 \leq 0$.

Plugging $A_2 \leq 0$ and Equation 83 into Equation 71,

$$\|\boldsymbol{w}_{s+1} - \boldsymbol{u}\|^2 \leq \|\boldsymbol{w}_s - \boldsymbol{u}\|^2 + 4\eta K \exp(\phi) F(\boldsymbol{u}_1) - 2\eta F(\boldsymbol{w}_s) \tag{91}$$

$$F(\boldsymbol{w}_s) \leq \frac{\|\boldsymbol{w}_s - \boldsymbol{u}\|^2 - \|\boldsymbol{w}_{s+1} - \boldsymbol{u}\|^2}{2\eta} + 2K \exp(\phi) F(\boldsymbol{u}_1), \tag{92}$$

and averaging over $s \in \{0, \ldots, r-1\}$ yields

$$\frac{1}{r} \sum_{s=0}^{r-1} F(\boldsymbol{w}_s) \leq \frac{\|\boldsymbol{w}_0 - \boldsymbol{u}\|^2 - \|\boldsymbol{w}_r - \boldsymbol{u}\|^2}{2\eta r} + 2K \exp(\phi) F(\boldsymbol{u}_1) \tag{93}$$

$$\leq \frac{\|\boldsymbol{w}_0 - (\boldsymbol{u}_1 + \boldsymbol{u}_2)\|^2}{2\eta r} + 2K \exp(\phi) F(\boldsymbol{u}_1) \tag{94}$$

$$\leq \frac{3}{2} \frac{\|\boldsymbol{w}_0\|^2 + \|\boldsymbol{u}_1\|^2 + \|\boldsymbol{u}_2\|^2}{\eta r} + 2K \exp(\phi) F(\boldsymbol{u}_1) \tag{95}$$

$$\leq \frac{3}{2} \frac{\|\boldsymbol{w}_0\|^2 + \lambda_1^2 + \lambda_2^2}{\eta r} + 2K \exp(\phi) F(\lambda_1 \boldsymbol{w}_*). \tag{96}$$

Recall that

$$F(\lambda_1 \boldsymbol{w}_*) = \frac{1}{Mn} \sum_{m=1}^{M} \sum_{i=1}^{n} \log(1 + \exp(-\lambda_1 \langle \boldsymbol{x}_i^m, \boldsymbol{w}_* \rangle)) \leq \log(1 + \exp(-\lambda_1 \gamma)) \overset{(i)}{\leq} \exp(-\lambda_1 \gamma), \tag{97}$$

where $(i)$ uses $\log(1 + x) \leq x$ for $x \geq 0$. So

$$\frac{1}{r} \sum_{s=0}^{r-1} F(\boldsymbol{w}_s) \leq \frac{3}{2} \frac{\|\boldsymbol{w}_0\|^2 + \lambda_1^2 + \lambda_2^2}{\eta r} + 2K \exp(\phi - \lambda_1 \gamma) \tag{98}$$

$$= \frac{3}{2} \frac{\|\boldsymbol{w}_0\|^2 + \lambda_1^2 + \lambda_2^2}{\eta r} + 2 \exp(\log K + \phi - \lambda_1 \gamma). \tag{99}$$

Here we choose $\lambda_1 = (\phi + \log(K + \eta K \gamma^2 r))/\gamma$. Finally, together with the previous choice of $\lambda_2 = \eta K/(2\gamma)$, we have

$$\frac{1}{r} \sum_{s=0}^{r-1} F(\boldsymbol{w}_s) \leq \frac{3\|\boldsymbol{w}_0\|^2}{2\eta r} + \frac{3(\phi^2 + \log^2(K + \eta K \gamma^2 r))}{\eta \gamma^2 r} + \frac{3\eta K^2}{8\gamma^2 r} + \frac{2}{1 + \eta \gamma^2 r} \tag{100}$$

$$\leq \frac{14\|\boldsymbol{w}_0\|^2}{\eta \gamma^4 r} + \frac{12\eta}{\gamma^4 r} + \frac{15 \log^2(K + \eta K \gamma^2 r)}{\eta \gamma^4 r} + \frac{3\eta K^2}{8\gamma^2 r} + \frac{26}{\eta \gamma^4 r} \tag{101}$$

$$\leq 26 \frac{\|\boldsymbol{w}_0\|^2 + 1 + \log^2(K + \eta K \gamma^2 r) + \eta^2 K^2}{\eta \gamma^4 r}. \tag{102}$$

$\square$

## A.2. Proof of Theorem 4.2

**Lemma A.3** (Restatement of Lemma 4.6). *If $F(\boldsymbol{w}_r) \leq 1/(4\eta M)$ for some $r \geq 0$, then $F_m(\boldsymbol{w}_{r,k}^m)$ is decreasing in $k$ for every $m$.*

*Proof.* Recall that for each $r, m$, the sequence of local steps $\{\boldsymbol{w}_{r,k}^m\}_k$ is generated by GD for a single-machine logistic regression problem. To show decrease of the objective, we use the modified descent inequality from Lemma 4.5.

We want to show that $F_m(\boldsymbol{w}_{r,k+1}^m) \leq F_m(\boldsymbol{w}_{r,k}^m)$ for every $k$. To do this, we prove $F_m(\boldsymbol{w}_{r,k}^m) \leq F_m(\boldsymbol{w}_r)$ by induction on $k$. Clearly it holds for $k = 0$, so suppose that it holds for some $0 \leq k < K$. Then

$$\|\boldsymbol{w}_{r,k+1}^m - \boldsymbol{w}_{r,k}^m\| = \eta \|\nabla F_m(\boldsymbol{w}_{r,k}^m)\| \overset{(i)}{\leq} \eta F_m(\boldsymbol{w}_{r,k}^m) \overset{(ii)}{\leq} \eta F_m(\boldsymbol{w}_r) \overset{(iii)}{\leq} 1/4, \tag{103}$$

where $(i)$ uses Lemma B.1, $(ii)$ uses the inductive hypothesis, and $(iii)$ uses $F_m(\boldsymbol{w}_r) \leq MF(\boldsymbol{w}_r) \leq 1/(4\eta)$. This bound on $\|\boldsymbol{w}_{r,k+1}^m - \boldsymbol{w}_{r,k}^m\|$ shows that the condition of Lemma 4.5 is satisfied, so

$$F_m(\boldsymbol{w}_{r,k+1}^m) - F_m(\boldsymbol{w}_{r,k}^m) \leq \langle \nabla F_m(\boldsymbol{w}_{r,k}^m), \boldsymbol{w}_{r,k+1}^m - \boldsymbol{w}_{r,k}^m \rangle + 4F_m(\boldsymbol{w}_{r,k}^m)\|\boldsymbol{w}_{r,k+1}^m - \boldsymbol{w}_{r,k}^m\|^2 \tag{104}$$

$$\leq -\eta \left\|\nabla F_m(\boldsymbol{w}_{r,k}^m)\right\|^2 + 4\eta^2 F_m(\boldsymbol{w}_{r,k}^m) \left\|\nabla F_m(\boldsymbol{w}_{r,k}^m)\right\|^2 \tag{105}$$

$$\leq -\eta \left(1 - 4\eta F_m(\boldsymbol{w}_{r,k}^m)\right) \left\|\nabla F_m(\boldsymbol{w}_{r,k}^m)\right\|^2 \tag{106}$$

$$\overset{(i)}{\leq} 0, \tag{107}$$

where $(i)$ uses the inductive hypothesis $F_m(\boldsymbol{w}_{r,k}^m) \leq F_m(\boldsymbol{w}_r) \leq 1/(4\eta)$. This completes the induction, so that $F_m(\boldsymbol{w}_{r,k}^m) \leq F_m(\boldsymbol{w}_r)$. Additionally, Equation 107 shows that $F_m(\boldsymbol{w}_{r,k}^m)$ is decreasing in $k$. $\qquad\square$

**Lemma A.4** (Restatement of Lemma 4.7). *If* $F(\boldsymbol{w}_r) \leq 1/(\eta KM)$ *for some* $r \geq 0$*, then* $\|\boldsymbol{w}_{r,k}^m - \boldsymbol{w}_r\| \leq 1$ *for every* $m \in [M], k \in \{0, \ldots, K-1\}$.

*Proof.* To bound the per-round movement $\|\boldsymbol{w}_{r,k}^m - \boldsymbol{w}_r\|$, we simply use the property $\|\nabla F_m(\boldsymbol{w})\| \leq F_m(\boldsymbol{w})$ from Lemma B.1, combined with the fact that the local loss is decreasing during the round from Lemma 4.6. Specifically,

$$\|\boldsymbol{w}_{r,k}^m - \boldsymbol{w}_r\| = \eta \left\|\sum_{t=0}^{k-1} \nabla F_m(\boldsymbol{w}_{r,t}^m)\right\| = \eta \sum_{t=0}^{k-1} \left\|\nabla F_m(\boldsymbol{w}_{r,t}^m)\right\| \tag{108}$$

$$\overset{(i)}{\leq} \eta \sum_{t=0}^{k-1} F_m(\boldsymbol{w}_{r,t}^m) \overset{(ii)}{\leq} \eta K F_m(\boldsymbol{w}_r) \overset{(iii)}{\leq} 1, \tag{109}$$

where $(i)$ uses Lemma B.1, $(ii)$ uses $F_m(\boldsymbol{w}_{r,t}^m) \leq F_m(\boldsymbol{w}_r)$ from Lemma 4.6, and $(iii)$ uses the condition $F_m(\boldsymbol{w}_r) \leq MF(\boldsymbol{w}_r) \leq 1/(\eta K)$. $\qquad\square$

**Lemma A.5** (Restatement of Lemma 4.8). *If* $F(\boldsymbol{w}_r) \leq \gamma/(70\eta KM)$*, then* $\|\boldsymbol{b}_r\| \leq \frac{1}{5}\|\nabla F(\boldsymbol{w}_r)\|$.

*Proof.* Our bound of $\|\boldsymbol{b}_r\|$ is essentially a direct calculation that leverages Lemmas B.4, B.1, and 4.6.

$$\|\boldsymbol{b}_r\| = \left\|\frac{1}{MK}\sum_{m=1}^{M}\sum_{k=0}^{K-1}(\nabla F_m(\boldsymbol{w}_{r,k}^m) - \nabla F_m(\boldsymbol{w}_r))\right\| \leq \frac{1}{MK}\sum_{m=1}^{M}\sum_{k=0}^{K-1}\left\|\nabla F_m(\boldsymbol{w}_{r,k}^m) - \nabla F_m(\boldsymbol{w}_r)\right\| \tag{110}$$

$$\overset{(i)}{\leq} \frac{1}{MK}\sum_{m=1}^{M}\sum_{k=0}^{K-1} 7F_m(\boldsymbol{w}_r)\|\boldsymbol{w}_{r,k}^m - \boldsymbol{w}_r\| = \frac{7}{MK}\sum_{m=1}^{M} F_m(\boldsymbol{w}_r)\sum_{k=0}^{K-1}\left\|\sum_{t=0}^{k-1}\eta \nabla F_m(\boldsymbol{w}_{r,t}^m)\right\| \tag{111}$$

$$\leq \frac{7\eta}{MK}\sum_{m=1}^{M} F_m(\boldsymbol{w}_r)\sum_{k=0}^{K-1}\sum_{t=0}^{k-1}\left\|\nabla F_m(\boldsymbol{w}_{r,t}^m)\right\| \overset{(ii)}{\leq} \frac{7\eta}{MK}\sum_{m=1}^{M} F_m(\boldsymbol{w}_r)\sum_{k=0}^{K-1}\sum_{t=0}^{k-1} F_m(\boldsymbol{w}_{r,t}^m) \tag{112}$$

$$\overset{(iii)}{\leq} \frac{7\eta K}{M}\sum_{m=1}^{M} F_m(\boldsymbol{w}_r)^2 \leq \frac{7\eta K}{M}\left(\sum_{m=1}^{M} F_m(\boldsymbol{w}_r)\right)^2 = 7\eta KMF(\boldsymbol{w}_r)^2 \tag{113}$$

$$\overset{(iv)}{\leq} \frac{\gamma}{10}F(\boldsymbol{w}_r) \overset{(v)}{\leq} \frac{1}{5}\|\nabla F(\boldsymbol{w}_r)\|, \tag{114}$$

where $(i)$ uses Lemma B.4 to bound the change in the local gradient during the round, $(ii)$ applies $\|\nabla F_m(\boldsymbol{w})\| \leq F_m(\boldsymbol{w})$ from Lemma B.1, $(iii)$ uses the fact that $F_m(\boldsymbol{w}_{r,t}^m)$ is decreasing in $t$ (Lemma 4.6), $(iv)$ uses the assumption $F(\boldsymbol{w}_r) \leq \gamma/(70\eta KM)$, and $(v)$ uses $F(\boldsymbol{w}) \leq \frac{2}{\gamma}\|\nabla F(\boldsymbol{w})\|$ from Lemma B.2. $\qquad\square$

**Lemma A.6.** *There exists some* $r \leq \tau$ *such that* $F(\boldsymbol{w}_r) \leq \frac{\gamma}{70\eta KM}$.

*Proof.* We use a potential function argument inspired by Lemma 9 of (Wu et al., 2024a). Similarly to our proof of Theorem 4.1, we have to account for the change in the local gradient $\nabla F_m(\boldsymbol{w}_{r,k}^m)$ during each round.

Define

$$G_m(\boldsymbol{w}) = \frac{1}{n} \sum_{i=1}^{n} |\ell'(\langle \boldsymbol{w}, \boldsymbol{x}_{m,i} \rangle)|, \tag{115}$$

and $G(\boldsymbol{w}) = \frac{1}{M} \sum_{m=1}^{M} G_m(\boldsymbol{w})$. Then for every $r \geq 0$,

$$\langle \boldsymbol{w}_{r+1}, \boldsymbol{w}_* \rangle = \langle \boldsymbol{w}_r, \boldsymbol{w}_* \rangle + \langle \boldsymbol{w}_{r+1} - \boldsymbol{w}_r, \boldsymbol{w}_* \rangle \tag{116}$$

$$= \langle \boldsymbol{w}_r, \boldsymbol{w}_* \rangle - \frac{\eta}{M} \sum_{m=1}^{M} \sum_{k=0}^{K-1} \langle \nabla F_m(\boldsymbol{w}_{r,k}), \boldsymbol{w}_* \rangle \tag{117}$$

$$= \langle \boldsymbol{w}_r, \boldsymbol{w}_* \rangle + \frac{\eta}{Mn} \sum_{m=1}^{M} \sum_{k=0}^{K-1} \sum_{i=1}^{n} |\ell'(\langle \boldsymbol{w}_{r,k}^m, \boldsymbol{x}_{m,i} \rangle)| \langle \boldsymbol{x}_{m,i}, \boldsymbol{w}_* \rangle \tag{118}$$

$$\geq \langle \boldsymbol{w}_r, \boldsymbol{w}_* \rangle + \frac{\eta\gamma}{Mn} \sum_{m=1}^{M} \sum_{k=0}^{K-1} \sum_{i=1}^{n} |\ell'(\langle \boldsymbol{w}_{r,k}^m, \boldsymbol{x}_{m,i} \rangle)| \tag{119}$$

$$= \langle \boldsymbol{w}_r, \boldsymbol{w}_* \rangle + \frac{\eta\gamma K}{Mn} \sum_{m=1}^{M} \sum_{i=1}^{n} \beta_{r,i}^m |\ell'(\langle \boldsymbol{w}_{r,k}^m, \boldsymbol{x}_{m,i} \rangle)| \langle \boldsymbol{x}_{m,i}, \boldsymbol{w}_* \rangle, \tag{120}$$

where $\beta_{r,i}^m := \frac{1}{K} \sum_{k=0}^{K-1} \frac{|\ell'(b_{r,i,k}^m)|}{|\ell'(b_{r,i}^m)|}$. We can lower bound $\beta_{r,i}^m \geq 1/K$ by ignoring all terms of the sum except the one corresponding to $k = 0$. This step was mentioned in our proof overview in Section 4. See Lemma B.7 for a discussion of the tightness of this step. $\beta_{r,i}^m \geq 1/K$ implies

$$\langle \boldsymbol{w}_{r+1}, \boldsymbol{w}_* \rangle \geq \langle \boldsymbol{w}_r, \boldsymbol{w}_* \rangle + \frac{\eta\gamma}{Mn} \sum_{m=1}^{M} \sum_{i=1}^{n} |\ell'(\langle \boldsymbol{w}_r, \boldsymbol{x}_{m,i} \rangle)| \tag{121}$$

$$= \langle \boldsymbol{w}_r, \boldsymbol{w}_* \rangle + \frac{\eta\gamma}{M} \sum_{m=1}^{M} G_m(\boldsymbol{w}_r) \tag{122}$$

$$= \langle \boldsymbol{w}_r, \boldsymbol{w}_* \rangle + \eta\gamma G(\boldsymbol{w}_r), \tag{123}$$

Rearraging and averaging over $r$,

$$\frac{1}{r} \sum_{s=0}^{r-1} G(\boldsymbol{w}_s) \leq \frac{\langle \boldsymbol{w}_r, \boldsymbol{w}_* \rangle - \langle \boldsymbol{w}_0, \boldsymbol{w}_* \rangle}{\eta\gamma r} \tag{124}$$

$$\leq \frac{\|\boldsymbol{w}_r - \boldsymbol{w}_0\|}{\eta\gamma r} \tag{125}$$

$$\overset{(i)}{\leq} \frac{2\gamma\|\boldsymbol{w}_0\| + \sqrt{2} + \eta + \log(1 + \eta\gamma^2 K r^2)}{\eta\gamma^2 r}, \tag{126}$$

where $(i)$ uses Lemma 4.4 together with $\|\boldsymbol{w}_r - \boldsymbol{w}_0\| \leq \|\boldsymbol{w}_r\| + \|\boldsymbol{w}_0\|$. Recall that $\psi = \min\left(\frac{\gamma}{140\eta KM}, \frac{1}{2Mn}\right)$; we want to the RHS of Equation 126 to be smaller than $\psi$. So we want

$$\psi \geq \frac{2\gamma\|\boldsymbol{w}_0\| + \sqrt{2} + \eta + \log(1 + \eta\gamma^2 K r^2)}{\eta\gamma^2 r} \tag{127}$$

$$r \geq \frac{2\gamma\|\boldsymbol{w}_0\| + \sqrt{2} + \eta + \log(1 + \eta\gamma^2 K r^2)}{\eta\gamma^2 \psi}. \tag{128}$$

Applying Lemma B.6 with

$$A = \frac{2\gamma\|\boldsymbol{w}_0\| + \sqrt{2} + \eta}{\eta\gamma^2 \psi}, \quad B = \frac{1}{\eta\gamma^2 \psi}, \quad C = \eta\gamma^2 K, \tag{129}$$

Equation 128 is satisfied when

$$r \geq \tau := \frac{1}{\eta\gamma^2\psi}\left(4\gamma\|\boldsymbol{w}_0\| + 2\sqrt{2} + 2\eta + \log\left(1 + \frac{\sqrt{K}}{\sqrt{\eta}\gamma\psi}\right)\right). \tag{130}$$

In particular, Equation 128 is satisfied with $r = \tau$. So, letting $r_0 = \arg\min_{0 \leq s < \tau} G(\boldsymbol{w}_s)$,

$$G(\boldsymbol{w}_{r_0}) \leq \frac{1}{\tau}\sum_{s=0}^{\tau-1} G(\boldsymbol{w}_s) \leq \psi. \tag{131}$$

We can now bound $F(\boldsymbol{w}_{r_0})$ in terms of $G(\boldsymbol{w}_{r_0})$. First, since $G(\boldsymbol{w}_{r_0}) \leq \frac{1}{2Mn}$, we have for each $m \in [M], i \in [n]$,

$$\frac{1}{Mn}|\ell'(\langle \boldsymbol{w}_{r_0}, \boldsymbol{x}_{m,i}\rangle)| \leq \frac{1}{Mn}\sum_{m=1}^{M}\sum_{i=1}^{n}|\ell'(\langle \boldsymbol{w}_{r_0}, \boldsymbol{x}_{m,i}\rangle)| = G(\boldsymbol{w}_{r_0}) \leq \frac{1}{2Mn}, \tag{132}$$

so

$$|\ell'(\langle \boldsymbol{w}_{r_0}, \boldsymbol{x}_{m,i}\rangle)| \leq \frac{1}{2} \tag{133}$$

$$\frac{1}{1 + \exp(\langle \boldsymbol{w}_{r_0}, \boldsymbol{x}_{m,i}\rangle)} \leq \frac{1}{2} \tag{134}$$

$$\langle \boldsymbol{w}_{r_0}, \boldsymbol{x}_{m,i}\rangle \geq 0, \tag{135}$$

so that every point is classified correctly by $\boldsymbol{w}_{r_0}$. Therefore

$$F(\boldsymbol{w}_{r_0}) = \frac{1}{Mn}\sum_{m=1}^{M}\sum_{i=1}^{n}\log(1 + \exp(-\langle \boldsymbol{w}_{r_0}, \boldsymbol{x}_{m,i}\rangle)) \tag{136}$$

$$\leq \frac{1}{Mn}\sum_{m=1}^{M}\sum_{i=1}^{n}\exp(-\langle \boldsymbol{w}_{r_0}, \boldsymbol{x}_{m,i}\rangle) \tag{137}$$

$$\overset{(i)}{\leq} \frac{1}{Mn}\sum_{m=1}^{M}\sum_{i=1}^{n}\frac{2}{1 + \exp(\langle \boldsymbol{w}_{r_0}, \boldsymbol{x}_{m,i}\rangle)} \tag{138}$$

$$\leq 2G(\boldsymbol{w}_{r_0}) \leq 2\psi = \min\left(\frac{\gamma}{70\eta KM}, \frac{1}{Mn}\right), \tag{139}$$

where $(i)$ uses $1 \leq \exp(\langle \boldsymbol{w}_{r_0}, \boldsymbol{x}_{m,i}\rangle)$. $\qquad\square$

**Theorem A.7** (Restatement of Theorem 4.2). *Denote $\psi = \min\left(\frac{\gamma}{140\eta KM}, \frac{1}{2Mn}\right)$ and*

$$\tau = \frac{4\gamma\|\boldsymbol{w}_0\| + 2\sqrt{2} + 2\eta + \log\left(1 + \frac{\sqrt{K}}{\sqrt{\eta}\gamma\psi}\right)}{\eta\gamma^2\psi}. \tag{140}$$

*For every $r \geq \tau$, Local GD satisfies*

$$F(\boldsymbol{w}_r) \leq \frac{16}{\eta\gamma^2 K(r - \tau)}. \tag{141}$$

*Proof.* The proof of this theorem has a similar structure as that of Lemma 4.6. When the loss $F(\boldsymbol{w}_s)$ is small, the total movement $\|\boldsymbol{w}_{s+1} - \boldsymbol{w}_s\|$ can be bounded (Lemma 4.7); when the movement is bounded, we can apply a modified descent inequality (Lemma 4.5), which shows decrease of the loss when $F(\boldsymbol{w}_s)$ is small. The main difference compared to Lemma 4.6 is that the update $\boldsymbol{w}_{s+1} - \boldsymbol{w}_s$ is not necessarily parallel with the gradient $\nabla F(\boldsymbol{w}_s)$. However, Lemma 4.8 shows that the magnitude of this bias is negligible compared to the magnitude of the gradient. Finally, Lemma 4.9 implies that the conditions of these lemmas (that $F(\boldsymbol{w}_r)$ is below some threshold) are met for some $r \leq \tau$. We execute this argument below.

By Lemma 4.9, there exists some $r_0 \leq \tau$ such that $F(\boldsymbol{w}_{r_0}) \leq \frac{\gamma}{70\eta KM}$. We will prove $F(\boldsymbol{w}_r) \leq F(\boldsymbol{w}_{r_0})$ for all $r \geq r_0$ by induction. Clearly it holds for $r = r_0$, so suppose it holds for some $r \geq r_0$. Notice that the condition $\|\boldsymbol{w}_{r+1} - \boldsymbol{w}_r\| \leq 1$ of Lemma 4.5 is satisfied, since

$$\|\boldsymbol{w}_{r+1} - \boldsymbol{w}_r\| = \left\| \frac{1}{M} \sum_{m=1}^{M} \boldsymbol{w}_{r,K}^m - \boldsymbol{w}_r \right\| \leq \frac{1}{M} \sum_{m=1}^{M} \|\boldsymbol{w}_{r,K}^m - \boldsymbol{w}_r\| \overset{(i)}{\leq} 1, \tag{142}$$

where $(i)$ uses Lemma 4.7. Recall that $\boldsymbol{w}_{r+1} - \boldsymbol{w}_r = -\eta K(\nabla F(\boldsymbol{w}_r) + \boldsymbol{b}_r)$. By applying Lemma 4.5:

$$F(\boldsymbol{w}_{r+1}) - F(\boldsymbol{w}_r) \tag{143}$$

$$\leq \langle \nabla F(\boldsymbol{w}_r), \boldsymbol{w}_{r+1} - \boldsymbol{w}_r \rangle + 4F(\boldsymbol{w}_r)\|\boldsymbol{w}_{r+1} - \boldsymbol{w}_r\|^2 \tag{144}$$

$$= -\eta K \langle \nabla F(\boldsymbol{w}_r), \nabla F(\boldsymbol{w}_r) + \boldsymbol{b}_r \rangle + 4\eta^2 K^2 F(\boldsymbol{w}_r) \|\nabla F(\boldsymbol{w}_r) + \boldsymbol{b}_r\|^2 \tag{145}$$

$$= -\eta K \|\nabla F(\boldsymbol{w}_r) + \boldsymbol{b}_r\|^2 + \eta K \langle \boldsymbol{b}_r, \nabla F(\boldsymbol{w}_r) + \boldsymbol{b}_r \rangle + 4\eta^2 K^2 F(\boldsymbol{w}_r) \|\nabla F(\boldsymbol{w}_r) + \boldsymbol{b}_r\|^2 \tag{146}$$

$$= -\eta K (1 - 4\eta K F(\boldsymbol{w}_r)) \|\nabla F(\boldsymbol{w}_r) + \boldsymbol{b}_r\|^2 + \eta K \langle \boldsymbol{b}_r, \nabla F(\boldsymbol{w}_r) + \boldsymbol{b}_r \rangle \tag{147}$$

$$\leq -\eta K (1 - 4\eta K F(\boldsymbol{w}_r)) \|\nabla F(\boldsymbol{w}_r) + \boldsymbol{b}_r\|^2 + \eta K \|\boldsymbol{b}_r\| \|\nabla F(\boldsymbol{w}_r) + \boldsymbol{b}_r\| \tag{148}$$

By Lemma 4.8, we have $\|\boldsymbol{b}_r\| \leq \frac{1}{5}\|\nabla F(\boldsymbol{w}_r)\|$. Therefore

$$\|\nabla F(\boldsymbol{w}_r) + \boldsymbol{b}_r\| \geq \|\nabla F(\boldsymbol{w}_r)\| - \|\boldsymbol{b}_r\| \geq 4\|\boldsymbol{b}_r\|, \tag{149}$$

so $\|\boldsymbol{b}_r\| \leq \|\nabla F(\boldsymbol{w}_r) + \boldsymbol{b}_r\|/4$. Plugging this back into Equation 148,

$$F(\boldsymbol{w}_{r+1}) - F(\boldsymbol{w}_r) \leq -\eta K \left( 1 - 4\eta K F(\boldsymbol{w}_r) - \frac{1}{4} \right) \|\nabla F(\boldsymbol{w}_r) + \boldsymbol{b}_r\|^2 \tag{150}$$

$$\overset{(i)}{\leq} -\frac{1}{2}\eta K \|\nabla F(\boldsymbol{w}_r) + \boldsymbol{b}_r\|^2 \tag{151}$$

$$\overset{(ii)}{\leq} -\frac{1}{4}\eta K \|\nabla F(\boldsymbol{w}_r)\|^2 \tag{152}$$

$$\overset{(iii)}{\leq} -\frac{1}{16}\eta\gamma^2 K F(\boldsymbol{w}_r)^2, \tag{153}$$

where $(i)$ uses the condition $F(\boldsymbol{w}_r) \leq \gamma/(70\eta KM)$, $(ii)$ uses

$$\|\nabla F(\boldsymbol{w}_r) + \boldsymbol{b}_r\| \geq \|\nabla F(\boldsymbol{w}_r)\| - \|\boldsymbol{b}_r\| \geq \frac{4}{5}\|\nabla F(\boldsymbol{w}_r)\|, \tag{154}$$

and $(iii)$ uses $\|\nabla F(\boldsymbol{w})\| \geq \frac{\gamma}{2}F(\boldsymbol{w})$ from Lemma B.2. Equation 148 completes the induction, so $F(\boldsymbol{w}_r) \leq F(\boldsymbol{w}_{r_0})$ for all $r \geq r_0$. Further, Equation 148 holds for all $r \geq r_0$, so we can unroll it to get an upper bound on $F(\boldsymbol{w}_r)$. Diving both sides of Equation 148 by $F(\boldsymbol{w}_r)F(\boldsymbol{w}_{r+1})$,

$$\frac{1}{F(\boldsymbol{w}_r)} - \frac{1}{F(\boldsymbol{w}_{r+1})} \leq -\frac{1}{16}\eta\gamma^2 K \frac{F(\boldsymbol{w}_r)}{F(\boldsymbol{w}_{r+1})} \tag{155}$$

$$\frac{1}{F(\boldsymbol{w}_{r+1})} \geq \frac{1}{F(\boldsymbol{w}_r)} + \frac{1}{16}\eta\gamma^2 K \frac{F(\boldsymbol{w}_r)}{F(\boldsymbol{w}_{r+1})} \tag{156}$$

$$\frac{1}{F(\boldsymbol{w}_{r+1})} \overset{(i)}{\geq} \frac{1}{F(\boldsymbol{w}_r)} + \frac{1}{16}\eta\gamma^2 K. \tag{157}$$

Unrolling from $r$ to $r_0$,

$$\frac{1}{F(\boldsymbol{w}_r)} \geq \frac{1}{F(\boldsymbol{w}_{r_0})} + \frac{1}{16}\eta\gamma^2 K(r - r_0) \geq \frac{1}{16}\eta\gamma^2 K(r - r_0), \tag{158}$$

so

$$F(\boldsymbol{w}_r) \leq \frac{16}{\eta\gamma^2 K(r - r_0)}. \tag{159}$$

Recall that $r_0 \leq \tau$, so $r - r_0 \geq r - \tau$, and finally

$$F(\boldsymbol{w}_r) \leq \frac{16}{\eta\gamma^2 K(r - \tau)}. \tag{160}$$

$\square$

## A.3. Proof of Corollary 4.3

**Corollary A.8** (Restatement of Corollary 4.3). *Suppose* $R \geq \widetilde{\Omega}\left(\max\left(\frac{Mn}{\gamma^2}, \frac{KM}{\gamma^3}\right)\right)$. *With* $\boldsymbol{w}_0 = \boldsymbol{0}$, $\eta \geq 1$, *and* $\eta K = \widetilde{\Theta}(\frac{\gamma^3 R}{M})$, *Local GD satisfies*

$$F(\boldsymbol{w}_R) \leq \widetilde{\mathcal{O}}\left(\frac{M}{\gamma^5 R^2}\right). \tag{161}$$

*Proof.* With our choices of $\boldsymbol{w}_0$, $\eta$, and $\eta K$, the transition time $\tau$ becomes

$$\tau = \frac{2\sqrt{2} + 2\eta + \log\left(1 + \frac{\sqrt{K}}{\sqrt{\eta}\gamma\psi}\right)}{\eta\gamma^2\psi} \tag{162}$$

$$= \widetilde{\mathcal{O}}\left(\frac{1 + \eta}{\eta\gamma^2\psi}\right) \tag{163}$$

$$\stackrel{(i)}{=} \widetilde{\mathcal{O}}\left(\frac{1}{\gamma^2\psi}\right) \tag{164}$$

$$\stackrel{(ii)}{=} \widetilde{\mathcal{O}}\left(\max\left(\frac{\eta KM}{\gamma^3}, \frac{Mn}{\gamma^2}\right)\right) \tag{165}$$

$$\stackrel{(iii)}{=} \widetilde{\mathcal{O}}\left(\max\left(R, \frac{Mn}{\gamma^2}\right)\right) \tag{166}$$

$$\stackrel{(iv)}{=} \widetilde{\mathcal{O}}(R), \tag{167}$$

where $(i)$ uses $\eta \geq 1$, $(ii)$ uses the definition of $\psi$, $(iii)$ uses the choice of $\eta K$, and $(iv)$ uses the condition

$$R \geq \widetilde{\Omega}\left(\frac{Mn}{\gamma^2}\right). \tag{168}$$

Therefore, we can ensure that $R \geq 2\tau$ with the appropriate choice of constant/logarithmic multiplicative factors on the RHS of Equation 168. Since $R \geq \tau$, Theorem 4.2 implies

$$F(\boldsymbol{w}_r) \leq \frac{16}{\eta\gamma^2 K(R - \tau)} \tag{169}$$

$$\stackrel{(i)}{\leq} \frac{32}{\eta\gamma^2 KR} \tag{170}$$

$$\stackrel{(ii)}{\leq} \widetilde{\mathcal{O}}\left(\frac{M}{\gamma^5 R^2}\right), \tag{171}$$

where $(i)$ uses $R - \tau \geq R/2$, since $R \geq 2\tau$, and $(ii)$ uses the choice $\eta K = \widetilde{\Theta}\left(\frac{\gamma^3 R}{M}\right)$. Note that the condition $R \geq \widetilde{\Omega}\left(\frac{KM}{\gamma^3}\right)$ is necessary to ensure that the choice $\eta K = \widetilde{\Theta}\left(\frac{\gamma^3 R}{M}\right)$ is compatible with the requirement $\eta \geq 1$. $\square$

## B. Auxiliary Lemmas

**Lemma B.1** (Lemma 25 from (Crawshaw et al., 2025)). *For every* $\boldsymbol{w} \in \mathbb{R}^d$,

$$\|\nabla F_m(\boldsymbol{w})\| \leq F_m(\boldsymbol{w}) \quad and \quad \|\nabla F(\boldsymbol{w})\| \leq F(\boldsymbol{w}). \tag{172}$$

**Lemma B.2** (Lemma 26 of (Crawshaw et al., 2025)). *If* $\boldsymbol{w} \in \mathbb{R}^d$ *such that* $\langle \boldsymbol{w}, \boldsymbol{x}_i^m \rangle \geq 0$ *for a given* $m \in [M]$ *and all* $i \in [n]$, *then*

$$\|\nabla F_m(\boldsymbol{w})\| \geq \frac{\gamma}{2} F_m(\boldsymbol{w}). \tag{173}$$

*Similarly, if* $\langle \boldsymbol{w}, \boldsymbol{w}_{m,i} \rangle \geq 0$ *for all* $m \in [M]$ *and all* $i \in [n]$, *then*

$$\|\nabla F(\boldsymbol{w})\| \geq \frac{\gamma}{2} F(\boldsymbol{w}). \tag{174}$$

**Lemma B.3** (Lemma 1 from (Crawshaw et al., 2025)). *For every $\boldsymbol{w}_1, \boldsymbol{w}_2 \in \mathbb{R}^d$,*

$$\|\nabla^2 F_m(\boldsymbol{w}_2)\| \le F_m(\boldsymbol{w}_1)\left(1 + \|\boldsymbol{w}_2 - \boldsymbol{w}_1\|\left(1 + \exp(\|\boldsymbol{w}_2 - \boldsymbol{w}_1\|^2)\left(1 + \frac{1}{2}\|\boldsymbol{w}_2 - \boldsymbol{w}_1\|^2\right)\right)\right). \tag{175}$$

**Lemma B.4.** *For $\boldsymbol{w}_1, \boldsymbol{w}_2 \in \mathbb{R}^d$, if $\|\boldsymbol{w}_1 - \boldsymbol{w}_2\| \le 1$, then*

$$\|\nabla F_m(\boldsymbol{w}_2) - \nabla F_m(\boldsymbol{w}_1)\| \le 7F_m(\boldsymbol{w}_1)\|\boldsymbol{w}_2 - \boldsymbol{w}_1\|. \tag{176}$$

*Proof.* The proof is a direct calculation, leveraging the upper bound of the objective's Hessian norm from Lemma B.3.

Let $\lambda = \|\boldsymbol{w}_2 - \boldsymbol{w}_1\|$ and $\boldsymbol{v} = \frac{\boldsymbol{w}_2 - \boldsymbol{w}_1}{\|\boldsymbol{w}_2 - \boldsymbol{w}_1\|}$. By the fundamental theorem of calculus,

$$\nabla F_m(\boldsymbol{w}_2) - \nabla F_m(\boldsymbol{w}_1) = \int_0^\lambda \nabla^2 F_m(\boldsymbol{w}_1 + t\boldsymbol{v})\boldsymbol{v}\, dt \tag{177}$$

$$\|\nabla F_m(\boldsymbol{w}_2) - \nabla F_m(\boldsymbol{w}_1)\| = \left\|\int_0^\lambda \nabla^2 F_m(\boldsymbol{w}_1 + t\boldsymbol{v})\boldsymbol{v}\, dt\right\| \tag{178}$$

$$\le \int_0^\lambda \left\|\nabla^2 F_m(\boldsymbol{w}_1 + t\boldsymbol{v})\boldsymbol{v}\right\|\, dt \tag{179}$$

$$\le \int_0^\lambda \left\|\nabla^2 F_m(\boldsymbol{w}_1 + t\boldsymbol{v})\right\|\, dt \tag{180}$$

$$\overset{(i)}{\le} \int_0^\lambda 7F_m(\boldsymbol{w}_1)\, dt \tag{181}$$

$$= 7F_m(\boldsymbol{w}_1)\lambda, \tag{182}$$

where $(i)$ uses Lemma B.3, noting that the condition $\|(\boldsymbol{w}_1 + t\boldsymbol{v}) - \boldsymbol{w}_1\| \le 1$ is satisfied by the assumption $\|\boldsymbol{w}_2 - \boldsymbol{w}_1\| \le 1$. $\square$

**Lemma B.5** (Restatement of Lemma 4.5). *For $\boldsymbol{w}, \boldsymbol{w}' \in \mathbb{R}^d$, if $\|\boldsymbol{w} - \boldsymbol{w}'\| \le 1$, then*

$$F_m(\boldsymbol{w}') \le F_m(\boldsymbol{w}) + \langle\nabla F_m(\boldsymbol{w}), \boldsymbol{w}' - \boldsymbol{w}\rangle + 4F_m(\boldsymbol{w})\|\boldsymbol{w}' - \boldsymbol{w}\|^2, \tag{183}$$

*and*

$$F(\boldsymbol{w}') \le F(\boldsymbol{w}) + \langle\nabla F(\boldsymbol{w}), \boldsymbol{w}' - \boldsymbol{w}\rangle + 4F(\boldsymbol{w})\|\boldsymbol{w}' - \boldsymbol{w}\|^2. \tag{184}$$

*Proof.* To prove this fact, we write $F_m$ as a second-order Taylor series centered at $\boldsymbol{w}$, then use Lemma B.3 to upper bound the quadratic term.

Let $\lambda = \|\boldsymbol{w}' - \boldsymbol{w}\|$ and $\boldsymbol{v} = \frac{\boldsymbol{w}' - \boldsymbol{w}}{\|\boldsymbol{w}' - \boldsymbol{w}\|}$. Then

$$F_m(\boldsymbol{w}') = F_m(\boldsymbol{w}) + \langle\nabla F_m(\boldsymbol{w}), \boldsymbol{w}' - \boldsymbol{w}\rangle + \underbrace{\int_0^\lambda (\lambda - t)\langle\boldsymbol{v}, \nabla^2 F_m(\boldsymbol{w} + t\boldsymbol{v})\boldsymbol{v}\rangle\, dt}_{Q}. \tag{185}$$

The quadratic term $Q$ can be bounded as follows:

$$Q \le \int_0^\lambda (\lambda - t)\|\boldsymbol{v}\|\left\|\nabla^2 F_m(\boldsymbol{w} + t\boldsymbol{v})\boldsymbol{v}\right\|\, dt \tag{186}$$

$$\le \int_0^\lambda (\lambda - t)\left\|\nabla^2 F_m(\boldsymbol{w} + t\boldsymbol{v})\right\|\, dt \tag{187}$$

$$\overset{(i)}{\le} 7F_m(\boldsymbol{w})\int_0^\lambda (\lambda - t)\, dt \tag{188}$$

$$= \frac{7}{2}F_m(\boldsymbol{w})\lambda^2, \tag{189}$$

where $(i)$ uses Lemma B.3 to bound $\|\nabla^2 F_m(\boldsymbol{w} + t\boldsymbol{v})\|$, using the condition that $\|(\boldsymbol{w} + t\boldsymbol{v}) - \boldsymbol{w}\| \leq \lambda \leq 1$. Plugging this into Equation 185 gives Equation 183, and averaging over $m \in [M]$ gives Equation 184. $\qquad\square$

**Lemma B.6.** *For $A, B, C \geq 0$, the inequality*

$$x \geq A + B \log(1 + Cx^2) \tag{190}$$

*is satisfied when*

$$x \geq 2A + B \log(1 + B\sqrt{C}). \tag{191}$$

*Proof.* Using concavity of $\sqrt{\cdot}$ and $\log$,

$$A + B \log(1 + Cx^2) = A + \frac{B}{2} \log(\sqrt{1 + Cx^2}) \tag{192}$$

$$\leq A + \frac{B}{2} \log(1 + \sqrt{C}x) \tag{193}$$

$$\leq A + \frac{B}{2}\left(\log(1 + B\sqrt{C}) + \frac{\sqrt{C}}{1 + B\sqrt{C}}(x - B)\right) \tag{194}$$

$$\leq A + \frac{B}{2}\left(\log(1 + B\sqrt{C}) + \frac{x}{B}\right) \tag{195}$$

$$= A + \frac{B}{2} \log(1 + B\sqrt{C}) + \frac{x}{2}. \tag{196}$$

So, to satisfy Equation 190, it suffices that

$$x \geq A + \frac{B}{2} \log(1 + B\sqrt{C}) + \frac{x}{2} \tag{197}$$

$$\frac{x}{2} \geq A + \frac{B}{2} \log(1 + B\sqrt{C}) \tag{198}$$

$$x \geq 2A + B \log(1 + B\sqrt{C}). \tag{199}$$

$\qquad\square$

An important part of the proofs of Theorem 4.1 and Lemma 4.9 is the lower bound

$$\beta_{r,i}^m := \frac{1}{K} \sum_{k=0}^{K-1} \frac{|\ell'(b_{r,i,k}^m)|}{|\ell'(b_{r,i}^m)|} \geq \frac{1}{K}, \tag{200}$$

which comes by ignoring all terms of the sum coming from $k > 0$. This may seem pessimistic, but the following lemma shows that for the case $n = 1$, this bound is tight up to logarithmic multiplicative factors for certain values of $\boldsymbol{w}_r$.

**Lemma B.7.** *Suppose $n = 1$ and $\boldsymbol{w}_r = \boldsymbol{0}$. Then $\beta_{r,i}^m \leq \mathcal{O}\left(\frac{1}{K} + \frac{1}{\eta\gamma^2 K} \log\left(1 + \eta\gamma^2 K\right)\right)$, and if additionally $\eta \geq 1$, then $\beta_{r,i}^m \leq \tilde{\mathcal{O}}\left(\frac{1}{K}\left(1 + \frac{1}{\gamma^2}\right)\right).$*

*Proof.* Since $n = 1$, we omit the index $i \in [n]$. We will also denote $\gamma_m = \|\boldsymbol{x}^m\|$. Recall that $\ell(z) = \log(1 + \exp(-z))$, so $|\ell'(z)| = \frac{1}{1+\exp(z)}$, and recall the definitions $b_r^m = \langle \boldsymbol{w}_r, \boldsymbol{x}^m \rangle$ and $b_{r,k}^m = \langle \boldsymbol{w}_{r,k}^m, \boldsymbol{x}^m \rangle$. Then we want to upper bound

$$\beta_r^m = \frac{1}{K} \sum_{k=0}^{K-1} \frac{1 + \exp(\langle \boldsymbol{w}_r, \boldsymbol{x}^m \rangle)}{1 + \exp(\langle \boldsymbol{w}_{r,k}^m, \boldsymbol{x}^m \rangle)}. \tag{201}$$

When $n = 1$, each local trajectory is relatively simple to analyze, since the updates $\boldsymbol{w}_{r,k+1}^m - \boldsymbol{w}_{r,k}^m$ are always parallel to $\boldsymbol{x}^m$. For this case, we will consider the gradient flow trajectory of $F_m$ initialized at $\boldsymbol{w}_r$. Since $n = 1$, the gradient flow has a convenient analytical form while also providing a lower bound for $b_{r,k}^m$, which will in turn give our upper bound for $\beta_r^m$.

Let $\widetilde{\boldsymbol{w}}_r^m : [0, \infty) \to \mathbb{R}^d$ be the gradient flow of $F_m$ initialized at $\boldsymbol{w}_r$, so that $\widetilde{\boldsymbol{w}}_r^m$ is the unique solution to

$$\frac{d}{dt}\widetilde{\boldsymbol{w}}_r^m(t) = -\eta \nabla F_m(\widetilde{\boldsymbol{w}}_r^m(t)) \quad \text{and} \quad \widetilde{\boldsymbol{w}}_r^m(0) = \boldsymbol{w}_r. \tag{202}$$

Then define $\widetilde{b}_r^m(t) = \langle \widetilde{\boldsymbol{w}}_r^m(t), \boldsymbol{x}^m \rangle$, so that

$$\frac{d}{dt}\widetilde{b}_r^m(t) = \left\langle \frac{d}{dt}\widetilde{\boldsymbol{w}}_r^m(t), \boldsymbol{x}^m \right\rangle \tag{203}$$

$$= -\eta \left\langle \nabla F_m(\widetilde{\boldsymbol{w}}_r^m(t)), \boldsymbol{x}^m \right\rangle \tag{204}$$

$$= -\eta \left\langle \frac{-\boldsymbol{x}_m}{1 + \exp(\langle \widetilde{\boldsymbol{w}}_r^m(t), \boldsymbol{x}^m \rangle)}, \boldsymbol{x}^m \right\rangle \tag{205}$$

$$= \frac{\eta \gamma_m^2}{1 + \exp(\widetilde{b}_r^m(t))}. \tag{206}$$

We claim that $\widetilde{b}_r^m(k) \leq b_{r,k}^m$, which we show by induction on $k$. Clearly it holds for $k = 0$, since $\widetilde{b}_r^m(0) = b_r^m = b_{r,0}^m$. So suppose it holds for some $k \geq 0$. If $\widetilde{b}_r^m(k+1) \leq b_{r,k}^m$, then we are done, since $b_{r,k+1}^m \geq b_{r,k}^m$. Otherwise, by the intermediate value theorem, there exists some $t_0 \in [k, k+1]$ such that $\widetilde{b}_r^m(t_0) = b_{r,k}^m$, so

$$\widetilde{b}_r^m(k+1) = \widetilde{b}_r^m(t_0) + \int_{t_0}^{k+1} \frac{d}{dt}\widetilde{b}_r^m(t)\, dt \tag{207}$$

$$= b_{r,k}^m + \eta \gamma_m^2 \int_{t_0}^{k+1} \frac{1}{1 + \exp(\widetilde{b}_r^m(t))}\, dt \tag{208}$$

$$\leq b_{r,k}^m + \eta \gamma_m^2 \int_{t_0}^{k+1} \frac{1}{1 + \exp(\widetilde{b}_r^m(t_0))}\, dt \tag{209}$$

$$= b_{r,k}^m + \eta \gamma_m^2 (k+1-t_0)\frac{1}{1 + \exp(b_{r,k+1}^m)} \tag{210}$$

$$\leq b_{r,k}^m + \eta \gamma_m^2 \frac{1}{1 + \exp(b_{r,k+1}^m)} \tag{211}$$

$$= b_{r,k+1}^m. \tag{212}$$

This completes the induction, so we know $\widetilde{b}_r^m(k) \leq b_{r,k}^m$ for all $k$. From Equation 201, this means

$$\beta_r^m \leq \frac{1 + \exp(b_r^m)}{K} \sum_{k=0}^{K-1} \frac{1}{1 + \exp(\widetilde{b}_r^m(k))}. \tag{213}$$

Also, we can directly solve the ODE in Equation 206 for $\widetilde{b}_r^m(t)$:

$$\frac{d}{dt}\widetilde{b}_r^m(t) = \frac{\eta \gamma_m^2}{1 + \exp(\widetilde{b}_r^m(t))} \tag{214}$$

$$(1 + \exp(\widetilde{b}_r^m(t)))\, d\widetilde{b}_r^m(t) = \eta \gamma_m^2 dt \tag{215}$$

$$\widetilde{b}_r^m(t) + \exp(\widetilde{b}_r^m(t)) = \eta \gamma_m^2 t + C \tag{216}$$

$$\widetilde{b}_r^m(t) + \exp(\widetilde{b}_r^m(t)) \overset{(i)}{=} \eta \gamma_m^2 t + b_r^m + \exp(b_r^m), \tag{217}$$

where $(i)$ comes from the initial condition $\widetilde{b}_r^m(0) = b_r^m$. For a fixed $t$, we use the substitutions $z = \exp(\widetilde{b}_r^m(t))$ and $b = \eta \gamma_m^2 t + b_r^m + \exp(b_r^m)$ to obtain

$$\log(z) + z = b \tag{218}$$

$$z \exp(z) = \exp(b) \tag{219}$$

$$z = W(\exp(b)), \tag{220}$$

where $W$ denotes the principal branch of the Lambert W function. So

$$\exp(\widetilde{b}_r^m(t)) = W(\exp(\eta\gamma_m^2 t + b_r^m + \exp(b_r^m))) \tag{221}$$

$$\widetilde{b}_r^m(t) = \log(W(\exp(\eta\gamma_m^2 t + b_r^m + \exp(b_r^m)))) \tag{222}$$

$$\widetilde{b}_r^m(t) = \log(W(\exp(1 + \eta\gamma_m^2 t))), \tag{223}$$

where we used the choice $\boldsymbol{w}_r = \boldsymbol{0} \implies b_r^m = 0$. Denoting $w = W(\exp(1 + \eta\gamma_m^2 t))$, we have by the definition of $W$

$$w\exp(w) = \exp(1 + \eta\gamma_m^2 t) \tag{224}$$

$$w + \log w = 1 + \eta\gamma_m^2 t \tag{225}$$

$$2w \overset{(i)}{\geq} 1 + \eta\gamma_m^2 t \tag{226}$$

$$w \geq \frac{1 + \eta\gamma_m^2 t}{2}, \tag{227}$$

where $(i)$ uses $\log w \leq w$. Plugging $w \geq \frac{1}{2}(1 + \eta\gamma_m^2 t)$ back into Equation 223 yields $\widetilde{b}_r^m(t) \geq \log(\frac{1}{2}(1 + \eta\gamma_m^2 t))$, and plugging this back into Equation 213 yields

$$\beta_r^m \leq \frac{1 + \exp(b_r^m)}{K} + \frac{1 + \exp(b_r^m)}{K} \sum_{k=1}^{K-1} \frac{1}{1 + \exp(\widetilde{b}_r^m(k))} \tag{228}$$

$$= \frac{2}{K} + \frac{2}{K} \sum_{k=1}^{K-1} \frac{1}{1 + \exp(\widetilde{b}_r^m(k))} \tag{229}$$

$$\leq \frac{2}{K} + \frac{4}{K} \sum_{k=1}^{K-1} \frac{1}{3 + \eta\gamma_m^2 k} \tag{230}$$

$$\leq \frac{2}{K} + \frac{4}{K} \int_0^{K-1} \frac{1}{3 + \eta\gamma_m^2 t} dt \tag{231}$$

$$= \frac{2}{K} + \frac{4}{\eta\gamma_m^2 K} \left[\log(3 + \eta\gamma_m^2 t)\right]_0^{K-1} \tag{232}$$

$$= \frac{2}{K} + \frac{4}{\eta\gamma_m^2 K} \log\left(1 + \frac{\eta\gamma_m^2(K-1)}{3}\right) \tag{233}$$

$$\leq \frac{2}{K} + \frac{4}{\eta\gamma_m^2 K} \log\left(1 + \frac{\eta\gamma_m^2 K}{3}\right) \tag{234}$$

$$\leq \frac{2}{K} + \frac{4}{\eta\gamma^2 K} \log\left(1 + \frac{\eta\gamma^2 K}{3}\right), \tag{235}$$

where the last line uses $\gamma_m = \|\boldsymbol{x}^m\| \geq \gamma$ together with the fact that $f(x) = \log(1 + x)/x$ is decreasing in $x$. $\qquad\square$

## C. Additional Experimental Details

The synthetic and MNIST datasets that we use for the experiments in Section 5 are described in full detail below.

### C.1. Synthetic Data

The synthetic dataset is a simple task with $M = 2$ clients and $n = 1$ data points per client, with $d = 2$ dimensional data. It was introduced by Crawshaw et al. (2025) with the goal of inducing conflict between the magnitude and direction of local client updates. The two data points $\boldsymbol{x}_1, \boldsymbol{x}_2$ are defined in terms of parameters $\delta, g$ as follows: $\boldsymbol{w}_1 = \gamma_1 \boldsymbol{w}_1^*$ and $\boldsymbol{w}_2 = \gamma_2 \boldsymbol{w}_2^*$, where

$$\boldsymbol{w}_1^* = \left(\frac{\delta}{\sqrt{1 + \delta^2}}, \frac{1}{\sqrt{1 + \delta^2}}\right) \tag{236}$$

$$\boldsymbol{w}_2^* = \left(\frac{\delta}{\sqrt{1 + \delta^2}}, -\frac{1}{\sqrt{1 + \delta^2}}\right), \tag{237}$$

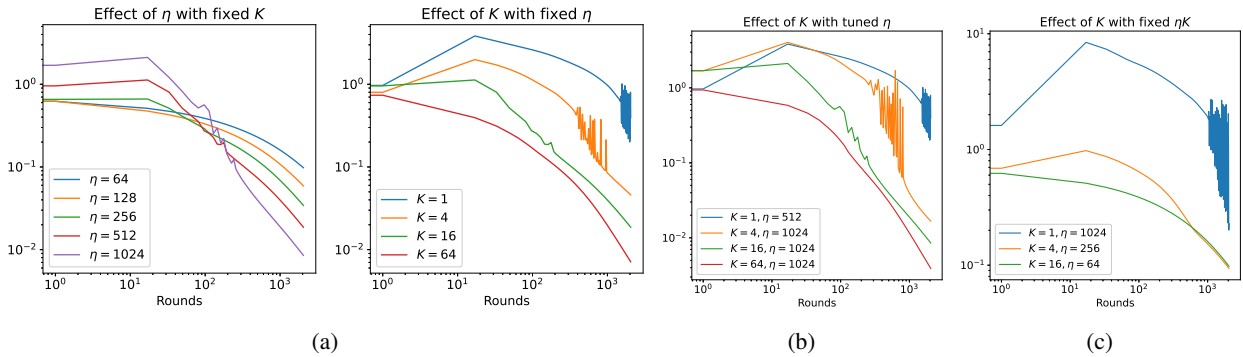

Figure 4: Train loss of Local GD (step size $\eta$, communication interval $K$) with the CIFAR-10 dataset. Overall, we observe that Local GD converges faster in the long run by choosing a larger step size/communication interval, despite unstable/slow optimization in early iterations. For $(a)$, we first fix $K = 16$ while varying $\eta$, then fix $\eta = 2^9$ while varying $K$.

and $\gamma_1 = 1, \gamma_2 = 1/g$. By choosing $\delta$ close to zero and $g$ with large magnitude, the two local objectives differ significantly in terms of gradient direction and magnitude. For our experiments, we use $\delta = 0.1$ and $g = 10$.

### C.2. MNIST

Similar to Wu et al. (2024a) and Crawshaw et al. (2025), we use a subset of MNIST data with binarized labels, and our implementation follows that of Crawshaw et al. (2025). First, we randomly select 1000 images from the MNIST dataset, which we then partition among the $M$ clients using a heterogeneity protocol that is common throughout the federated learning literature (Karimireddy et al., 2020). Specifically, for a data similarity parameter $s \in [0, 100]$, the $s\%$ of the data is allocated to an "iid pool", which is randomly shuffled, and a "non-iid pool", which is sorted by label. When sorting the non-iid pool, we sort according to the 10-way digit label. We then split the iid pool into $M$ equally sized subsets, and similarly split the non-iid pool into $M$ equally sized subsets (keeping the sorted order), and each client's local dataset is comprised of one subset of the iid pool together with one subset of the non-iid pool. In this way, the local datasets have different proportions of each digit. If $s = 100$, then the 1000 images are allocated uniformly at random to different clients, and if $s = 0$, then the clients will have nearly disjoint sets of digits in their local datasets. Finally, after images have been allocated to clients, we replace each image's label with the parity of its depicted digit. For our experiments, we set $M = 5$ and $s = 50$. For all images, the pixel values initially fall into the range $[0, 255]$; we normalize the data by subtracting 127 from each pixel, then dividing all pixels by the same scaling factor to ensure that $\max_{m,i} \|\boldsymbol{x}_i^m\| = 1$.

## D. Additional Experimental Results

### D.1. CIFAR-10 Experiments

In this section, we provide additional experiments on the CIFAR-10 dataset, using similar protocols as in Section 5. For these experiments, we vary the step size $\eta \in \{2^6, 2^7, \ldots, 2^{10}\}$, and other details of the setup exactly match those of our MNIST experiments (see Section C.2), including the number of communication rounds $R$, the heterogeneity procedure, number of clients $M$, number of samples per client $n$, data similarity parameter $s$, data normalization procedure, and choice of interval $K \in \{1, 4, 16, 64\}$. Note that we used step sizes between $2^6$ and $2^{10}$, since smaller choices led to very slow, very stable convergence and larger choices led to overflow.

The results can be seen in Figure 4. For these additional experiments, we used the same evaluation protocol as in Section 5: Figures 4(a) corresponds to Q1 and Figure 1, Figure 4(b) corresponds to Q2 and Figure 2, and Figure 4(c) corresponds to Q3 and Figure 3.

The results on CIFAR-10 further support our theoretical findings. In Figure 4(a), larger step sizes/communication intervals lead to faster convergence in the long run, despite the resulting slow/unstable convergence in early iterations. In Figure 4(b), we can see that a larger communication interval $K$ leads to faster convergence when $\eta$ is tuned to $K$. The results in Figure 4(c) are similar to the MNIST results in Figure 3: when $\eta K$ is constant, $K = 1$ is less stable and slower than other choices of $K$, and all other choices have roughly the same final loss. These results strengthen the evidence that our

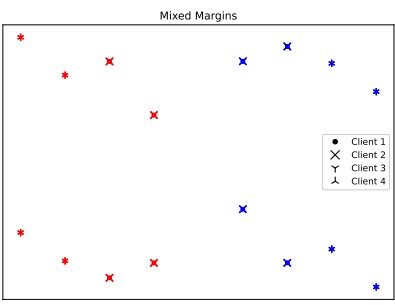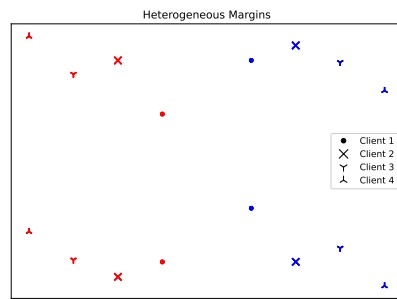

Figure 5: Three splits of a synthetic dataset. Binary labels are shown in red/blue, and client indices for each data point are shown with markers. Note that some data points are contained by multiple clients, which is shown with overlapping markers. In the homogeneous split (left), all clients have the same data, so they all have the same local margins. For mixed (middle), two clients have local margin $\gamma$, and two clients have local margin $3\gamma$. For heterogeneous (right), all four clients have different local margins. Note that the combined dataset of all four clients is the same for all three splits.

theoretical findings accurately describe the behavior of Local GD in practice.

### D.2. Margin Heterogeneity

While our theoretical analysis makes no assumption about data heterogeneity (it applies to any linearly separable dataset), the question remains whether the convergence rate can be improved with a more fine-grained analysis that considers the local margins $\gamma_m := \max_{\boldsymbol{w} \in \mathbb{R}^d, \|\boldsymbol{w}\|=1} \min_{(x,y) \in D_m} y \langle \boldsymbol{w}, \boldsymbol{w} \rangle$ instead of the global margin $\gamma$ alone. We investigate this question with a controlled synthetic dataset, by changing the local margins $\gamma_m$ while preserving the global dataset.

This synthetic dataset has $M = 4$ clients with a total of 16 data points. The dataset can be split among the four clients in three different ways to create either homogeneous, partially homogeneous (i.e. mixed), or heterogeneous margins among clients, which are shown in Figure 5. Note that $\|\boldsymbol{x}_i^m\| \leq 1$ for every data point, so that $H \leq 1/4$, similarly with the datasets of Section 5. Also, the global dataset (and therefore $\gamma$) is the same for all three splits. Our theory provides the same convergence rate upper bound for all three splits, and we verify this prediction by evaluating Local GD with various hyperparameters on the three splits. Results are shown in Figure 6.

The left subplots of Figure 6 show that the losses for each split are slightly different in early iterations, but quickly become nearly identical. The right subplots show that all three splits satisfy $\eta \gamma^2 K r \cdot F(\boldsymbol{w}_r) \to 1$ as $r$ increases, so that the asymptotic convergence rate is unaffected by heterogeneity in the local margins. This behavior is consistent across choices of $\eta$ and $K$. These results align with our theoretical prediction that the convergence rate of Local GD depends on properties of the global dataset, rather than how that dataset is allocated among clients.

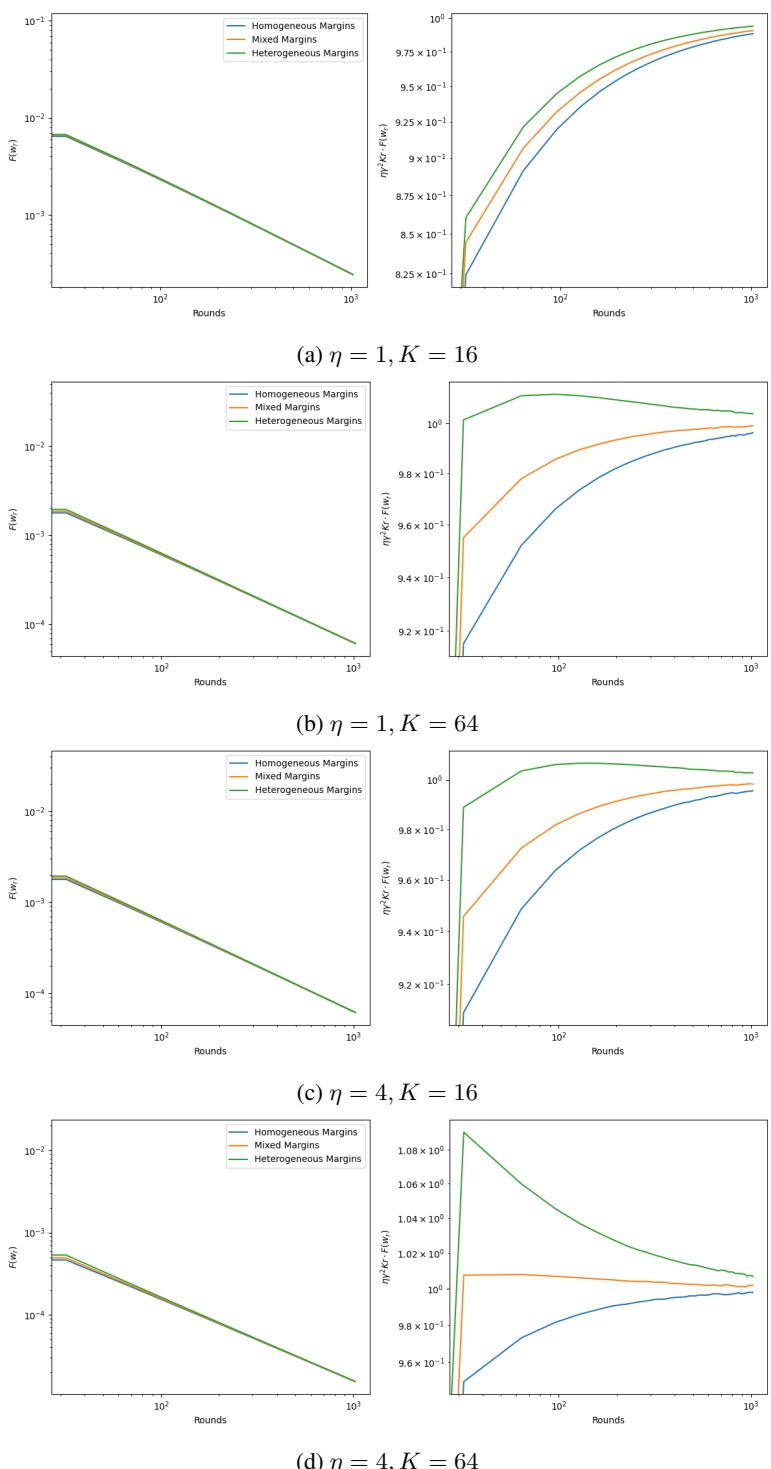

(a) $\eta = 1, K = 16$

(b) $\eta = 1, K = 64$

(c) $\eta = 4, K = 16$

(d) $\eta = 4, K = 64$

Figure 6: Results of Local GD on three splits of the synthetic dataset pictured in Figure 5. The right subplots show the asymptotic rate as the number of iterations goes to $\infty$, similarly to Figures 1(b) and 1(d) of (Wu et al., 2024a).

