# OpenReview forum: "Constant Stepsize Local GD for Logistic Regression: Acceleration by Instability"
_ICML.cc/2025/Conference — ICML 2025 poster_

### Official Review · Reviewer_d7rc · 2025-03-13

**Overall Recommendation:** 3

**Summary:**

The paper establishes improved convergence rates for local gradient descent in the context of distributed logistic regression with separable data. This improvement is attained by employing significantly larger step sizes than those typically used for general smooth loss functions.

# Update after rebuttal

In the rebuttal, the authors discussed my concerns regarding the tightness of the bound and the technical challenges in the proofs. Although it is not clear whether the bound is tight with respect to $\gamma$, I am leaning towards acceptance.

**Claims And Evidence:**

Yes. All of the theoretical results are proved in the paper.

**Essential References Not Discussed:**

To the best of my knowledge, the authors have discussed all relevant related works.

**Experimental Designs Or Analyses:**

The experiments focus primarily on synthetic and MNIST datasets.

**Methods And Evaluation Criteria:**

Yes. The paper is mostly theoretical and contains only basic numerical evaluations.

**Other Comments Or Suggestions:**

N.A

**Other Strengths And Weaknesses:**

### Strengths

1. The paper is well-written and clearly structured. The authors provide a detailed analysis and offer valuable intuition to aid the reader's understanding.

2. The authors achieve the best-known convergence rates for distributed logistic regression with separable data.

3. The paper demonstrates that the technique introduced by Wu et al. (2024) for obtaining improved bounds with extremely large step sizes is also applicable in the distributed setting.

### Weaknesses

1. From a technical standpoint, the analysis largely builds on the work of Wu et al. (2024) for the non-distributed setting.

2. It remains unclear whether the bound provided by the authors is tight for the specific problem they consider.

**Questions For Authors:**

1. In the authors' opinion, is the additional factor $ M $ and $1/\gamma $ in the bound, compared to the non-distributed setting, tight?

2. What are the main challenges in the analysis beyond those already addressed in Wu et al. (2024) for non-distributed gradient descent?

**Relation To Broader Scientific Literature:**

The paper improves the convergence rates established in (Woodworth et al., 2020b), (Koloskova et al., 2020), and (Crawshaw et al., 2025) for the problem of distributed logistic regression with separable data.

From a technical perspective, much of the methodology builds upon the approach of (Wu et al., 2024a) for gradient descent in the non-distributed setting.

**Theoretical Claims:**

I checked the correctness of the proofs in the main text.

---

> ### Author Rebuttal · Authors · 2025-03-30
>
> Thank you for your feedback on our submission. Below we have responded to the comments in your review.
>
> 1. **Tightness in terms of $M$ and $\gamma$.** This is an interesting question. For $M$, the current dependence may be tight, but of course we cannot know for sure without a lower bound. Here we offer some speculation about the tightness. Our $M$ dependence arises from the requirement that $F(w_r) \leq O(\gamma/(\eta KM))$ of Lemmas 4.6-4.8, which guarantees that Local GD has entered the stable phase. The factor of $1/M$ is needed in that requirement to ensure that $F_m(w_r)$ is small for every single client $m$, since $F(w) \leq C$ implies that $F_m(w) \leq MC$. Our proof of Theorem A.7 then guarantees stable descent when each client's loss $F_m(w_r)$ is small. For the question of tightness, it comes down to whether the inequality $F_m(w) \leq MC$ is tight. While this inequality may appear pessimistic, it is actually tight in the case that all client losses are close to zero except one, and we have experimentally observed cases like this even in simple settings like $M=2$. In those instances, there is a strong oscillatory behavior, where at each iteration one client loss is close to zero while the other is large. It is possible that such behavior also occurs with larger $M$. If so, then the previous inequality is sometimes tight, and in that case the $M$ dependence may be unavoidable. Based on this preliminary evidence, we guess that the complexity may have some unavoidable dependence on $M$, but again, there is no way to be totally rigorous without providing a lower bound.
>
>     For $\gamma$, we do believe that the dependence can be improved, although we guess that it requires some additional analysis which is outside the scope of the current paper. As we pointed out in our submission, the dependence in terms of $\gamma$ is slightly worse than the single-machine case, which suggests that some tightening is possible. However, we believe that it requires more fine-grained knowledge of the trajectory of Local GD.  In particular, we may be able to tighten the dependence on $\gamma$ if we know the implicit bias of Local GD. We leave this kind of fine-grained trajectory analysis for future work, and for now we just focus on the convergence rate in terms of the number of communication rounds $R$. If you want to know more technical details about the origin and possible solution to this issue, please see our response to reviewer f231.
>
> 2. **Main challenges in the analysis.** The key challenge of the analysis is to bound the time to transition to the stable phase *even under local updates*. If $K=1$, then this can already be performed with the potential function argument of (Wu et al, 2024a).  With local updates ($K > 1$), it is not immediately clear whether the same potential function can be used, and if it can be used, whether it decreases at the same rate as in the single-machine case. The key insight to bridge this gap is to decompose the round update $w_{r+1} - w_r$ into the contributions of each individual data point, and to upper and lower bound the contribution of each data point. This allows us to relate gradient potential of Local GD to that of GD, and the argument is executed in Lemma 4.9 (Lemma A.6 in the Appendix). This same decomposition is also the key step to prove Theorem 4.1. Our Section 4.2 also gives an overview of this decomposition and how it is used in the proof for both the stable and unstable phase.

---

### Official Review · Reviewer_f231 · 2025-03-14

**Overall Recommendation:** 3

**Summary:**

This paper studies local gradient descent (GD) for logistic regression with separable data in a distributed setting. Building on prior work by [Wu et al., 2024], which showed that a large stepsize improves optimization efficiency in a single-machine setting, this work extends the analysis to multiple machines with multiple local GD steps. Similar to [Wu et al., 2024], the authors demonstrate that a large stepsize benefits the optimization process.

**Claims And Evidence:**

See below.

**Essential References Not Discussed:**

See below.

**Experimental Designs Or Analyses:**

See below.

**Methods And Evaluation Criteria:**

See below.

**Other Comments Or Suggestions:**

Overall, I find the paper interesting and well written, but there are a few areas for improvement:

1. **Comparison to the Single-Machine Case:**
   When $K=1$ or $M=1$, the problem reduces to the single-machine setting. However, the obtained bound appears worse than that in [Wu et al., 2024] by some factors of $\gamma$. While this issue is briefly mentioned after Corollary 4.3 and in Section 6, it would be valuable to further explore its technical origins. Specifically, identifying which step in the analysis introduces this looseness would provide greater clarity. It seems unlikely that the current bound is tight in terms of $\gamma$.

2. **Exploration of Local Steps ($K>1$) in Theory:**
   The discussion on the benefits of local steps is interesting. In the context of logistic regression with separable data, more local steps help local GD enter the stable regime faster, enabling the use of a larger stepsize for a given optimization budget. While this phenomenon is demonstrated in simulations, providing theoretical support would significantly enhance the significance of the paper.

3. **Handling of Heterogeneous Data Distributions:**
   The current analysis assumes a uniform margin $\gamma$ across all devices, which simplifies the problem but overlooks data heterogeneity. When performing local steps, local GD is influenced only by the local margin instead of the global margin, which can be much larger than the global margin $\gamma$. A more fine-grained local analysis could reveal additional benefits of local steps and offer new insights into distributed/federated optimization.

I think the paper would be much stronger if the above three issues could be resolved properly. In its current form, I think the paper is on the borderline case. At least the first issue should be addressed during the rebuttal.

**Other Strengths And Weaknesses:**

See below.

**Questions For Authors:**

See above.

**Relation To Broader Scientific Literature:**

See below.

**Theoretical Claims:**

See below.

---

> ### Author Rebuttal · Authors · 2025-03-30
>
> Thank you for your insightful comments. Below we have responded to the points in your review.
>
> 1. **Comparison to the single-machine case.** The issue of $\gamma$ dependence stems from the gradient bias $b_r$ in Lemma A.5. Notice that other conditions for entering the stable phase (Lemma A.3, Lemma A.4) only require $F(w_r) \leq O(1/(\eta KM))$, whereas Lemma A.5 requires $F(w_r) \leq O(\gamma/(\eta KM))$. This extra factor of $\gamma$ needed to bound $\lVert b_r \rVert$ creates the worse dependence on $\gamma$ compared with the single-machine case. Note that the gradient bias results from taking local steps before averaging, so it does not appear when $K=1$ or $M=1$. We will also note that while we may not have a tight dependence on $\gamma$ here, it is also not clear that it would be the same as in Wu et al in the case $K,M>1$.
>
>     Technically, the requirement $F(w_r) \leq O(\gamma/(\eta KM))$ might be weakened, but only with a more fine-grained trajectory analysis. First, note that the requirement on $F(w_r)$ is used in Equation (114) of Lemma A.5, for the inequality marked $(iv)$. The need for the factor of $\gamma$ arises from $(v)$, where we apply $F(w) \leq \frac{1}{\gamma} \lVert \nabla F(w) \rVert$ (from Lemma B.2). The additional factor of $\gamma$ is needed to cancel the $1/\gamma$ from Lemma B.2. Now, if we had a stronger bound in Lemma B.2 --- say $F(w) \leq \lVert \nabla F(w) \rVert$ --- then we could remove the extra $\gamma$ factor. Unfortunately, the bound $F(w) \leq \lVert \nabla F(w) \rVert$ does not hold for all $w$, but it does hold, for example, when $w = t w_*$, where $t$ is a large scalar. In summary, we might improve the gamma dependence if we knew that Local GD converges near the max-margin solution. Unfortunately, this kind of implicit bias result would require more analysis and is outside the scope of this paper. Even in the single-machine case, the implicit bias of GD for logistic regression is unknown when the learning rate scales linearly in the number of iterations (Wu et al, 2024a). Investigating the implicit bias would require a significant amount of work which we leave as a future direction.
>
> 2. **Exploration of local steps in theory.** The question of the benefit of local steps is a fundamental problem in distributed optimization, which we discussed thoroughly in Section 6. We acknowledge that our results do not show an improvement from local steps, though we would like to point out that the same can be said of nearly all results in this line of work (see (Woodworth et al 2020b) and (Patel et al, 2024) for thorough discussions of the literature). Even for a fixed setting like logistic regression, proving the benefit of local steps is nontrivial and is outside the scope of our single paper. We plan to address this fundamental question in follow up works.
>
> 3. **Handling of heterogeneous data distributions.** First, we should clarify we make no restrictive assumptions about data heterogeneity. Our analysis handles any heterogeneous dataset that is linearly separable. $\gamma$ is the maximum margin of the combined dataset, but we do not assume that it is the maximum margin of every local dataset.
>
>     The question remains whether the complexity can be improved with a fine-grained analysis that considers the local margins instead of just the global one. We believe the answer is no: changes in the local margins alone cannot improve the asymptotic convergence rate, as we explain below.
>
>     Theoretically, our analysis shows that once the loss is small, Local GD is essentially GD on the global dataset with some small gradient bias (Lemma A.5 + Theorem A.7). So after this threshold, the convergence rate is determined by properties of the global dataset. The local margins could affect the time it takes to reach this threshold, but after the threshold, the convergence rate is determined by the global margin $\gamma$. So the local margins do not affect the convergence rate as the number of iterations goes to $\infty$.
>
>     Experimentally, we designed a dataset to test Local GD while varying the local margins. We use three ways of splitting the global dataset which creates either homogeneous, partially heterogeneous, or totally heterogeneous local margins. The dataset is visualized in Figure 2 of https://anonymous.4open.science/r/25_icml_rebuttal-FEC8/, and the results are shown in Figure 3. The left subplots of Figure 3 show that the losses for each split are slightly different in early iterations, but quickly become nearly identical. The right subplots show that all three splits satisfy $\eta \gamma^2 Kr \cdot F(w_r) \rightarrow 1$ as $r$ increases, so that the asymptotic convergence rate is unaffected by heterogeneity in the local margins. This behavior is consistent across choices of $\eta$ and $K$.

---

> > ### Comment · Reviewer_f231 · 2025-04-03
> >
> > Thank you for your response.
> >
> > Regarding 1, I think it will be useful to include these discussions in the revision so that experts could see where the extra $\gamma$ factor comes from.
> >
> > Regarding 2, it might be worth expending the related discussions in the paper to better highlight the open issue.
> >
> > Regarding 3, I was trying to suggest that considering local margin might help improve the phase transition bound, which might lead to some benefit of local step (sorry my wording might not be clear in the first place). But this is very much an open problem. Thank you for the additional simulations.
> >
> > I will raise my score to 3 to indicate that I am still near the borderline but leaning towards acceptance.

---

> > > ### Author Response · Authors · 2025-04-04
> > >
> > > Thank you for your response. We can definitely include the content from points 1 and 2 in the revised edition. Your point 3 is very interesting, and we agree that the local margins may play a role in the transition time. We hope to address this point in future work.

---

### Official Review · Reviewer_nvcT · 2025-03-14

**Overall Recommendation:** 3

**Summary:**

The authors demonstrate that Local GD for distributed logistic regression converges for any step size $\eta$ > 0 and any communication interval K ≥ 1. Experimental results on both synthetic and real-world data support the theoretical finding that acceleration is possible by permitting nonmonotonic decreases in the objective.

**Claims And Evidence:**

All theorems and corollaries are claimed clearly and provided with proofs.

**Essential References Not Discussed:**

No.

**Experimental Designs Or Analyses:**

The designs of experiments make sense. But still lack the performance with respect to different setups.

**Methods And Evaluation Criteria:**

Yes.

**Other Comments Or Suggestions:**

Each equation should be labeled with a single number, rather than being numbered on every line.

**Other Strengths And Weaknesses:**

The theoretical proofs are very solid and clear, and the future study mentioned in the paper is interesting.

**Questions For Authors:**

1. Have you tested more datasets to verify your method?

**Relation To Broader Scientific Literature:**

The authors adapt techniques from the analysis of GD with large step sizes for single-machine logistic regression to demonstrate that Local GD for distributed logistic regression converges for any step size $\eta$ > 0 and any communication interval K ≥ 1. Experimental results on both synthetic and real-world data support the theoretical finding that acceleration is possible by permitting nonmonotonic decreases in the objective.

**Theoretical Claims:**

No.

---

> ### Author Rebuttal · Authors · 2025-03-30
>
> Thank you for your efforts in the review process. Below we have responded to your comments about additional experimental setups.
>
> 1. **Additional experimental setups**. In the review, you asked "Have you tested more datasets to verify your method?". First, we would like to clarify that the goal of our paper is not to propose a new method that achieves the best possible performance, but rather to develop theory that accurately explains the practical behavior of fundamental machine learning algorithms, i.e. distributed gradient descent for binary classification with the logistic loss. With that in mind, the purpose of the experiments in our main submission is to experimentally verify our theoretical findings, that optimization can indeed be accelerated by choosing a large step size/communication interval.
>
>     To complement these results, we have added further experiments with the CIFAR-10 dataset, and we see largely the same behavior as predicted by our theory. The results can be seen in Figure 1 of https://anonymous.4open.science/r/25_icml_rebuttal-FEC8/. For these additional experiments, we used the same evaluation protocol as the experiments of the main submission: Figures 1(a), 1(b) correspond to Figure 1 of the main submission, Figure 1(c) corresponds to Figure 2 of the main submission, and Figure 1(d) corresponds to Figure 3 of the main submission.
>
>     These CIFAR-10 experiments further support our theoretical findings. In Figure 1(a) of the linked PDF, larger step sizes/communication intervals lead to faster convergence in the long run, despite the resulting slow/unstable convergence in early iterations. In Figure 1(b), we can see that a larger communication interval $K$ leads to faster convergence when $\eta$ is tuned to $K$. The results in Figure 1(c) are similar to the MNIST results in Figure 3 of the main body: when $\eta K$ is constant, $K=1$ is less stable and slower than other choices of $K$, and all other choices have roughly the same final loss. These results strengthen the evidence that our theoretical findings accurately describe the behavior of Local GD in practice.
>
>     Lastly, we added another experiment (in response to reviewer f231) with a synthetic dataset to investigate the effect of heterogeneity among the margins of the local datasets. Please see our response to reviewer f231 for more information on this additional experiment, whose results can also be found in the linked PDF.

---

### Decision · Program_Chairs · 2025-05-01

**Decision:**

Accept (poster)

**Comment:**

The paper considers the problem of distributed optimization and attacks a fundamental bottleleck in this setting. In particular the paper considers local GD, where each server executes K GD steps on local data before communication. In this setting the learning rate was known to necessarily be decayed at a rate of 1/K but this was to ensure continuous decay. The paper leverages recent work in the case of logisitic regression of linearly separable data with margin that showed that after a burn-in period of potential loss increase, the loss eventually decreases and at a faster rate achieving acceleration. The paper establishes a similar phenomenon in the setting of local GD, showing that an LR >> 1/K can be used and they show a burn-in period of loss increase and then eventual decrease and acceleration.

Overall there was a solid reviewer author discussion on this paper and the reviewers all unanimously recommended the paper to be accepted. Several great suggestions were made by the reviewers regarding the underlying theory, discussions along the lines of which will strongly embellish the paper. Given the previous work the paper is somewhat on the edge in terms of novelty and some of the points discussed the discussion phase if resolved will put the paper very clearly above the bar. I am recommending a borderline accept in the current form aligning with the views of the reviewers.